# Photothermia at the nanoscale induces ferroptosis via nanoparticle degradation

Alexandre Fromain[1,4], Jose Efrain Perez [1,4], Aurore Van de Walle[1], Yoann Lalatonne [2,3] & Claire Wilhelm [1] ✉

The Fe(II)-induced ferroptotic cell death pathway is an asset in cancer therapy, yet it calls into question the biocompatibility of magnetic nanoparticles. In the latter, Fe(II) is sequestered within the crystal structure and is released only upon nanoparticle degradation, a transition that is not well understood. Here, we dissect the chemical environment necessary for nanoparticle degradation and subsequent Fe(II) release. Importantly, temperature acts as an accelerator of the process and can be triggered remotely by laser-mediated photothermal conversion, as evidenced by the loss of the nanoparticles' magnetic fingerprint. Remarkably, the local hot-spot temperature generated at the nanoscale can be measured in operando, in the vicinity of each nanoparticle, by comparing the photothermal-induced nanoparticle degradation patterns with those of global heating. Further, remote photothermal irradiation accelerates degradation inside cancer cells in a tumor spheroid model, with efficiency correlating with the endocytosis progression state of the nanoparticles. High-throughput imaging quantification of $Fe^{2+}$ release, ROS generation, lipid peroxidation and cell death at the spheroid level confirm the synergistic thermo-ferroptotic therapy due to the photothermal degradation at the nanoparticle level.

Since their introduction into the field, iron oxide magnetic nanoparticles certainly remain one of the most used materials in nanomedicine[1,2]. Initially developed as contrast agents for magnetic resonance imaging, they were next proposed as multifunctional therapeutic agents for drug delivery[3-8] and anticancer thermal therapy[9-12]. The nanoparticles' iron composition benefits from this element's essential role in oxygen transport, enzymatic functions, organ growth, and survival. This translates to an excellent biocompatibility that connects their ultimate transformation with their bio-assimilation within an endogenous iron-related metabolism, such as the ferritin storage pathway, acting as a regulator of the iron byproducts[13-23]. In addition to the natural biocompatibility of Fe(III), its bio-assimilation is thus established thanks to the mechanisms of maintenance of iron homeostasis at the cell level.

Nevertheless, iron can be toxic in its unbound, redox-active state Fe(II). Consequently, the role of iron in disease emerged with ferroptosis revealed as a form of cell death mediated by Fe(II)-triggered oxidative damage[24]. On the one hand, iron-interacting molecules that induce ferroptosis (e.g., inhibitors of GPX4) have been envisaged as anticancer strategy[25]. On the other hand, ferroptosis can be detrimental in neurodegenerative diseases, which has prompted the introduction of iron-regulating drugs[26]. Within this context, the biocompatibility of iron oxide nanoparticles in the organism must be revisited. In these nanoparticles, Fe(II) is sequestered within the crystal structure, and for it to become active, the nanoparticles have to be degraded. Eliciting such a response could then lead to ferroptosis[27,28], with the iron-dependent accumulation of reactive oxygen species by the Fenton reaction culminating in cell damage.

[1]Laboratoire Physico Chimie Curie, PCC, CNRS UMR168, Institut Curie, Sorbonne University, PSL University, 75005 Paris, France. [2]Université Sorbonne Paris Nord, Université Paris Cité, Laboratory for Vascular Translational Science, LVTS, INSERM, UMR 1148, F- 93017 Bobigny, France. [3]Département de Biophysique et de Médecine Nucléaire, Assistance Publique-Hôpitaux de Paris, Hôpital Avicenne, F- 93009 Bobigny, France. [4]These authors contributed equally: Alexandre Fromain, Jose Efrain Perez. ✉e-mail: claire.wilhelm@cnrs.fr

Ferroptosis activation thus recently appeared as a novel way to use iron-based nanoparticles in cancer therapy[29,30], adding up to their therapeutic assets. It involved iron complexes and nanozymes[31–33], or the use of nanoparticles to serve as Fe(II) reservoirs upon intracellular degradation[34–41].

A critical crossroads then becomes apparent: one path leading to safe Fe(III)-content nanoparticles for use in regenerative medicine, and the other to unsafe Fe(II)-content nanoparticles triggering biological damage, but with potential for cancer therapy. The latter is in line with the use of magnetic nanoparticles in hyperthermia therapy in oncology, which consists in the elevation of the tumor temperature to induce apoptosis. There are two main nanoparticle-based thermal therapies: magnetic hyperthermia[42–44] and photothermal therapy[45,46], based on the use of a magnetic field or photothermal irradiation via laser, respectively[47,48]. Notably, iron oxide nanoparticles are potent agents for both modalities, with photothermal conversion being the most efficient treatment at low doses of nanoparticles in the cellular environment[49,50]. Contrary to plasmonic nanoparticles, their photoheating capacity does not significantly depend on aggregation (e.g., in endosomes) or coating.

This work has the goal to investigate the interplay between photothermal therapy with iron oxide nanoparticles, their degradation and subsequent ferroptosis induction. The first essential milestone was to elucidate the physical, chemical, and biological conditions that control nanoparticle degradation. Moreover, the proof of concept that degradation may be caused by a global temperature increase was yet to be evidenced. After dissecting this effect in a lysosomal-like solution, it was further demonstrated that a punctual temperature increase by photothermal irradiation can also act as a remote accelerator of nanoparticle degradation. Besides, the degradation extent mirrors the hot-spot temperature induction in the vicinity of each nanoparticle, which can be quantified as a function of power density, giving an unprecedented in operando measure of temperature at the nanoscale. Next, this degradation acceleration efficiency is evidenced to be dependent on the rate of nanoparticle internalization, from early to late endosomes, in an in cellulo spheroid model. Lastly, using a high-throughput spheroid model, it is evidenced that the induced nanoparticle degradation by photothermia at the nanoscale culminates in a synergistic ferroptotic-photothermal therapy response.

## Results and discussion

### A magnetic sensor to follow nanoparticle degradation

Iron oxide superparamagnetic nanoparticles with an average diameter of $9.6 \pm 2.1$ nm were synthesized through the sol–gel microwave-assisted method[51] and coated with citrate as a stabilizing agent (Supplementary Fig. 1)[52]. The study is based on the exploitation of the magnetometric signal of the nanoparticles as a direct representation of their integrity in order to monitor their degradation under different environments and stimulations in real time. In this work, we used a custom bench-top magnetic sensor device (Fig. 1A) to have a rapid and effective magnetometric quantification[53]. The device's output is a dynamic signal based on the non-linearity of the magnetization of the nanoparticles, which is proportional to the quantity of magnetic nanoparticles. Briefly, nanoparticles are exposed to a two-frequency alternating magnetic field varying between two amplitudes. It allows a dynamic monitoring of nanoparticle magnetization without any sample preparation. The output signal, expressed in arbitrary units, can be converted into magnetic moment (emu), as both measurements showed a very strong correlation ($r^2 > 0.99$) (Fig. 1B). While standard magnetometry methods such as vibrating-sample magnetometer (VSM) provide direct information on the magnetic moment of a material as a function of temperature, field, and crystal orientation, the measurement time can be decreased only down to 5 min per sample at room temperature, and biological samples need to be fixed. Herein,

with the magnetic sensor, the measurement time is 10 s, a real convenience when testing time-dependent degradation effects on many samples, and to work with live biological samples. Yet, it only provides a measure of the quantity of magnetic nanoparticles in the sample, with principle introduced in more detail in Supplementary Fig. 2 and in previous work[53]. Besides, calibration (shown in Fig. 1B) requires the use of VSM.

### Environment required for nanoparticle degradation

Using the magnetic sensor, we established the kinetics of degradation, highlighting the role of pH, iron chelation and global temperature. We used a solution that mimics the acidic pH of lysosomes, with added citrate as an iron chelating agent[54]. We dispersed nanoparticles in water at [Fe] = 2 mM at pH = 4.5, with citrate concentrations ranging from 0 to 20 mM. Results show a clear influence of the citrate concentration on degradation, with maximum degradation reached after 22 h at room temperature in the [Citrate] = 20 mM condition (Fig. 1C). An increase in global temperature to 60 °C significantly accelerated the degradation process, reaching similar levels to the room temperature condition in just under 2 h. We next subjected nanoparticle solutions at a twofold [Citrate]:[Fe] ratio to decreasing pH conditions at room temperature, with maximum nanoparticle degradation reached at around pH = 4.5 (Fig. 1D). Iron chelation and acidic pH thus govern the rate of nanoparticle degradation. Lastly, we then subjected the same solution to increasing temperatures over the course of two hours (Fig. 1E). Remarkably, the degradation process is dependent on and accelerated by increases in temperature, additionally verified by the change in color density of the nanoparticles solution as they degrade (Fig. 1F).

### Photothermia-induced degradation at the nanoscale

The next step was to take advantage of the nanoparticles' photothermal conversion capacity to remotely increase the temperature. Figure 2A and B shows, respectively, the Infra-Red (IR) images and temperature curves of the heating of nanoparticles solutions in water (non-degrading environment) subjected to increasing laser power densities, revealing a rapid temperature increase, up to 80 °C at 2.4 W/cm², yet without changes in the nanoparticles' crystallinity (as evidenced with Fourier transform infrared spectroscopy in Supplementary Fig. 3). The same measurements were done in lysosomal-like degradation medium, with a comparable initial temperature increase, but then experiencing a steady loss of temperature after a few minutes of irradiation (Fig. 2C). Such decrease is certainly an indicator of nanoparticle degradation, but not a quantitative one, as heat transfer and saturation complicates the relationship between the number of heating sources and temperature. We also investigated any possible heating effect of the released Fe²⁺ ions only, by doing the same series of measurements with ferrous ascorbate at the same iron concentration. Supplementary Fig. 4 shows the temperature curves, and reveals a low heating efficacy, less than 15% of the one of nanoparticles. The extent of magnetic degradation was then measured quantitatively with the magnetic sensor after 10 min (Fig. 2D) and 30 min (Fig. 2E) of laser application for each power density. An almost complete nanoparticle degradation was reached in 30 min for power densities over 2 W/cm².

Remarkably, the photothermal-induced degradation outperforms the degradability of global heating. This evidences the fact that the local temperature at the nanoparticle level after laser exposure must significantly differ from the global, surrounding temperature (Fig. 2F). Consequently, the difference in $\Delta T = T_{local} - T_{global}$ is the direct temperature imprint of the hot-spot surrounding each nanoparticle, derived from the degradation measurements upon laser exposure relative to that of global heating, through the polynomial regression $T_{global} = f(\text{Degradation}_{global})$. For each degradation measurement at one laser power density, $\text{Degradation}_{laser}$, we can then

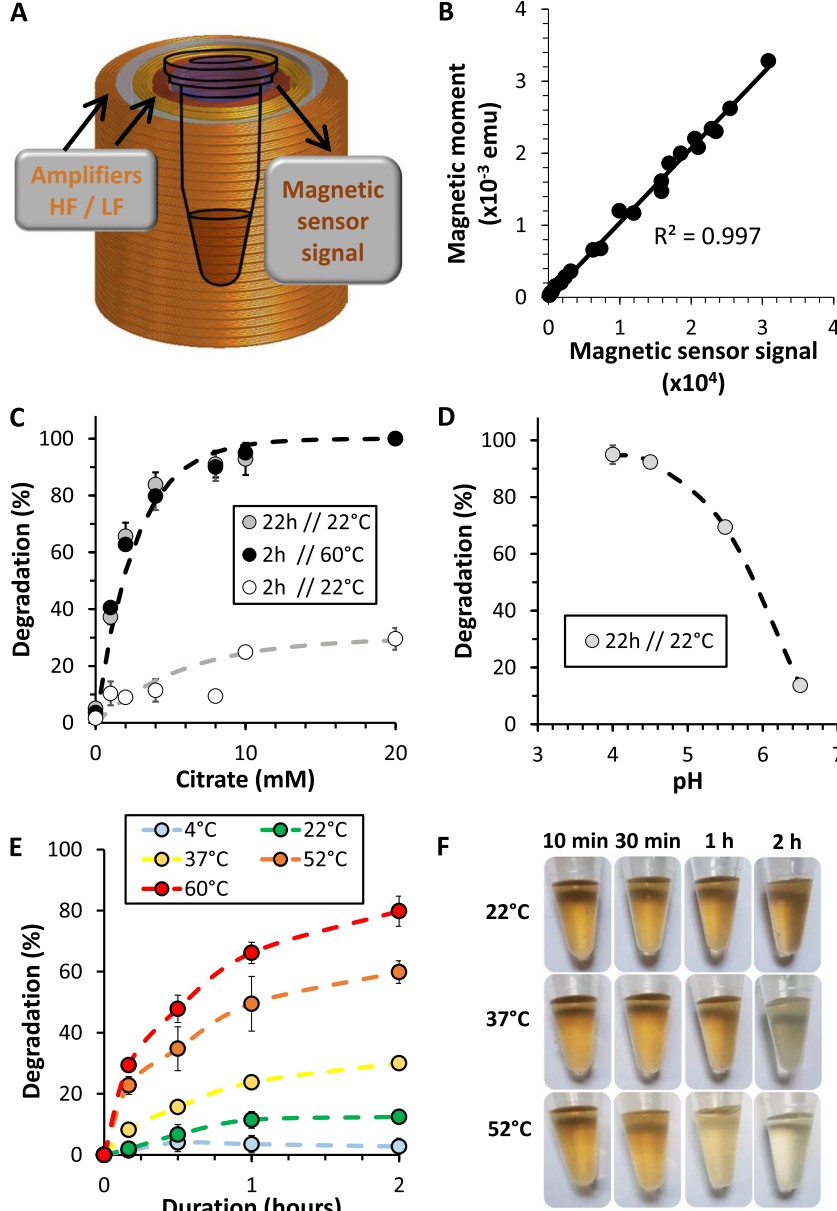

**Fig. 1 | Magnetism as a fingerprint of the nanoparticle degradation process in a lysosomal-like solution. A** Schematization of the magnetic sensor, made of two independent coils that generate a two-frequency magnetic field. HF high frequency, LF low frequency. **B** Correlation between the magnetic sensor signal and the magnetic moment obtained via VSM. **C** Degradation rate (%) in aqueous dispersion at pH = 4.5 with nanoparticles at [Fe] = 2 mM and with varying amounts of iron chelating agent at 22 °C and 60 °C ($n \geq 3$). **D** Effect of pH on degradation for the [Fe] = 2 mM and [Citrate] = 4 mM condition ($n \geq 3$). **E** Nanoparticle degradation rate in aqueous dispersion at pH = 4.5, [Fe] = 2 mM, and [Citrate] = 4 mM, subjected to increasing temperatures ($n \geq 3$). **F** Panel of previous nanoparticle solutions showing a change in color density following degradation. $n$ values represent the number of independent replicas, where data are presented as mean ± SD.

calculate the corresponding global temperature so that $\Delta T =$ f(Degradation$_{laser}$) − $T_{laser}$. Figure 2G shows that the calculated hot-spot temperature increases linearly with the laser power density, reaching up to 20 °C more than the global temperature at high power.

To the best of our knowledge, this is the first showcase of the nanoparticle hot-spot measurement using photothermia with iron oxide nanoparticles. Other attempts have been made using molecular thermometers[55–57], thermo-responsive polymers[58], enzymatic activity of fluorescent proteins[44], or X-ray absorption spectroscopy[59,60], to assess the hotspots around plasmonic or magnetic nanoparticles during photothermal treatment or magnetic hyperthermia, respectively[59,61–66]. In the case of iron oxide-mediated magnetic hyperthermia, high-temperature gradients were reported a

few nm away from the magnetic nanoparticle surface[62,63,66], reaching temperature differences up to 40−80 °C. In the case of photothermia mediated by plasmonic gold-based nanoparticles, thermosensitive fluorescent probes allowed measuring temperature gradients up to multiple folds the one of the global temperature[67–69]. Such use of independent nano-thermometers in the vicinity of the nanoparticles yet does not provide a direct nanoparticle-linked measurement, except for dual near-infrared (NIR) laser heaters and thermal sensors such as rare earth emmitters[70], or encapsulation of gold nanorods within a polymer matrix melting[71]. Another attractive option is to exploit X-ray absorption as a nanothermometric technique to monitor the internal local temperature within the nanoparticle core by direct changes in its structure, under photothermal heating[59], magnetic hyperthermia[60], or microwave irradiation[72], all

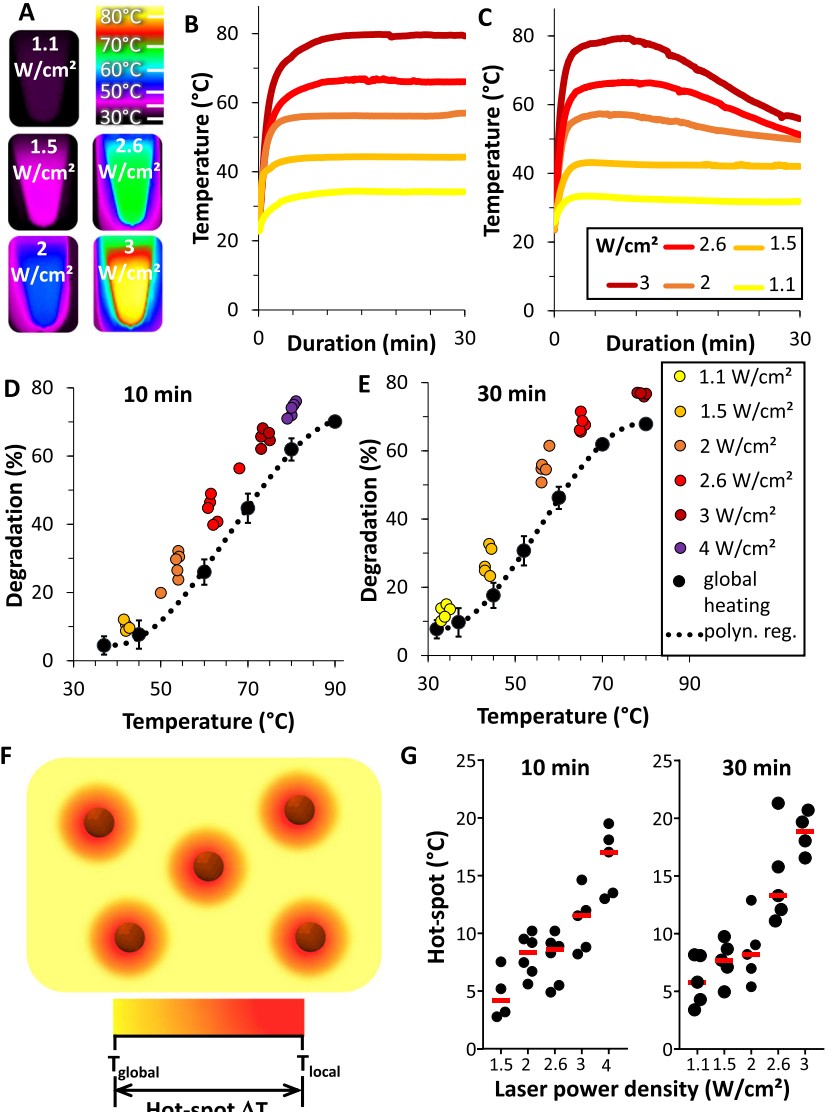

**Fig. 2 | Photothermia at the nanoscale deduced from the global temperature-mediated degradation. A** IR images of nanoparticles at [Fe] = 2 mM in water exposed to increasing 808 nm laser power densities. **B**, **C** Corresponding temperature increase curves in water and in degradation medium with [Cit] = 4 mM and pH = 4.5, respectively. **D**, **E** Degradation as a function of global heating (black curve) or upon laser at different power densities, after 10 and 30 min, respectively ($n \geq 3$). **F** Schematization of local photothermal heating compared to global heating.

**G** Nanoparticle hot-spot temperature as a function of laser power density ($n = 5$). This was defined as the local nanoparticle temperature induced by photothermal irradiation, equal to the controlled global temperature needed to reach the same degradation rate after photothermal heating, calculated from the polynomial regression (polyn. reg.) of (**D**, **E**). $n$ values represent the number of independent replicas, where data are presented as mean ± SD.

converging on a high-temperature increase at the level of the nanoparticle core compared to the outer medium. Of importance, these sophisticated techniques do not take into account the impact that the local temperature could have on the nanoparticles' functions. Here, we believe we provide an active, real working local temperature determination that reflects the degradation state reached by the nanoparticles after photothermal application.

### Controlled location after nanoparticle internalization in a tumor spheroid model

The demonstration that it is possible to trigger nanoparticle degradation after a photothermal-induced hot-spot temperature increase opens the opportunity for possible anticancer therapeutic use through iron release and ferroptosis. In order to further investigate iron release by degradation in the cellular environment, we established a 3D glioblastoma spheroid model of 400,000 cells matured for two days. Spheroids were incubated with nanoparticles (Fig. 3A) at [Fe] = 10 mM

with [Citrate] = 20 mM for 30 min at 4 °C. The nanoparticles interact with the cells at the outer periphery of the spheroids, as depicted visually (Fig. 3B) and by Prussian Blue iron staining (Fig. 3C). The 4 °C condition inhibits the internalization process, localizing all nanoparticles at the cell membrane interface. The nanoparticle uptake pathway was then kick-started by introducing spheroids back to a temperature of 37 °C. The incubation times at this temperature were then varied: for 30-min incubation, nanoparticles are mostly localized at the cell membrane or in early endosomes, at 2 h they can be considered to have reached endosomes, and after a 4-h incubation, they localize in late endosomes and lysosomes (Fig. 3D). To verify these assumptions, iron staining of spheroid tissue histological sections (Fig. 3E–G) and transmission electron microscopy (TEM) imaging (Fig. 3H–J) were performed. After 30 min, iron was detected at the cell membrane, on the edge of the spheroid (Fig. 3E, H), and nanoparticles were clearly identified at the cell membrane and in early endosomes near the cell membrane (Fig. 3F, I). On the other hand, spheroids

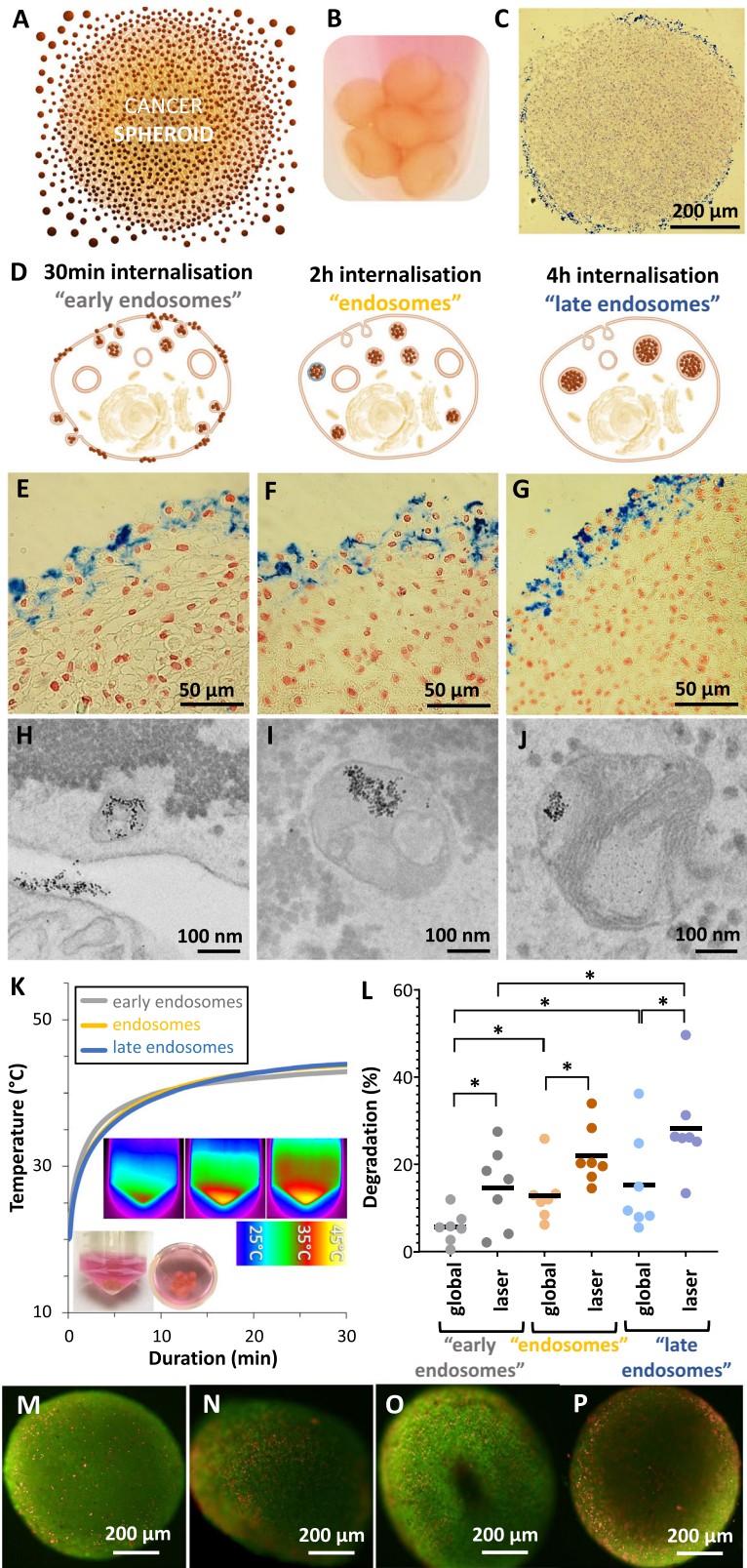

incubated at four hours showed a deeper cellular endosomal localization (Fig. 3J) as pointed by punctual iron signal (Fig. 3G), with no nanoparticles observed near the cell membrane. Additional histological section and TEM images provided in Supplementary Figs. 5–8 confirm these nanoparticle localizations.

We then proceeded to use the established spheroid endocytosis model to study photothermal-induced nanoparticle degradation.

Supplementary Fig. 9 shows the experimental setup for laser application to cancer spheroids models. The spheroids presented no significant variation in total amount of iron between conditions and after the heating treatment, evidencing no nanoparticle loss (Supplementary Fig. 10). Following the previous incubation method, the spheroids were irradiated at a power density of 2 W/cm² in order to reach the temperature range for hyperthermia (42–45 °C). The heating profiles

**Fig. 3 | Nanoparticle degradation in the intracellular environment of a tumor spheroid model. A** Schematization of nanoparticles interaction with the spheroids outer layer. **B** Image of spheroids after the initial incubation in a solution of nanoparticles with [Fe] = 2 mM and [Citrate] = 10 mM at 4 °C. **C** Prussian Blue showing iron in blue, evidencing the presence of nanoparticles at the spheroid periphery. **D** Schematization of nanoparticle uptake after incubation of the spheroids for 30 min (early endosomes), 2 h (endosomes) and 4 h (late endosomes) at 37 °C. **E–G** Prussian Blue staining of spheroids under the same conditions. Endosomal progression can be identified by a stronger, punctual blue signal within the cells. **H–J** TEM imaging for the same conditions, evidencing a similar progression of nanoparticle localization. **K** Spheroids heating profile for the three nanoparticles localization conditions after application of a laser power density of 2 W/cm² for 30 min. Insets show corresponding thermal IR camera images. **L** Average nanoparticle degradation after 30 min heating at 42 °C depending on their intracellular location (*n* > 3). **M–P** Composite fluorescence image illustrating the cytotoxicity of the heating treatment in spheroids. Images show merging of live green and dead red for the early endosomes (**M, N**) and late endosomes (**O, P**) after global (**M, O**) or laser (**N, P**) heating. *n* values represent the number of independent replicas. Unpaired two-tailed Student's *t* test was used to evaluate statistical significance, where **P* < 0.05.

were similar for all conditions, reaching saturation at 43–44 °C, highlighting that the endocytosis state does not impact photothermal conversion efficiency (Fig. 3K). In situ spheroid degradation was next evaluated under global and laser application, with the latter showing a consistently higher degradation regardless of the location of the nanoparticles in the tumor spheroid model (Fig. 3L). Remarkably, the later stages of the endocytosis pathway were found to be more favorable to degradation, at a 30% maximum on average. TEM images of spheroids at early and late endosomal nanoparticle internalization and subsequent laser exposure are shown in Supplementary Figs. 11 and 12, respectively. It resembles the spontaneous degradation patterns of iron oxide nanoparticles observed in other works, without stimulation[73,74]. It is not possible to detect degraded nanoparticles on these images. However, some intact nanoparticles can still be detected, along with other structures such as iron-loaded ferritin, or dark fingerprints within endosomes, both likely a result of iron oxide nanoparticles degradation. In order to still have a quantitative evaluation of the laser-mediated effect on nanoparticles at the nanoscale, laser was applied either in water (pH = 7, no degradation), or in degrading medium (pH = 4.5 with citrate), and samples were observed with TEM, with results shown in Supplementary Fig. 13. While there are absolutely no changes in the nanoparticles size after laser application in water, a small, yet significant decrease is seen in a degrading medium.

Cell viability of the spheroids was next assessed using a live/dead cell assay (Fig. 3M–P). Fluorescence imaging reveals a higher presence of dead cells in laser conditions, and higher for late endosomes than for early ones. Fluorescence quantification confirmed these results, as shown in Supplementary Fig. 14, with cell viability for global heating found at 86% and 74%, compared to 64% and 61% for photothermal application for early and late endosomes, respectively. Such a difference in cell death reached for the same global heating is likely to be the result of the higher degradation of the nanoparticles that could induce a cytotoxic response through the release of $Fe^{2+}$ and Fenton reaction.

## The role of nanoparticle Fe(II)/Fe(III) composition in hot-spot generation and ferroptosis induction

To further explore the role of the release of $Fe^{2+}$ from the nanoparticles, we next modified their composition through an oxidation process that converts their magnetite composition ($Fe_3O_4$) into maghemite ($\gamma$-$Fe_2O_3$), as confirmed with Fourier transform infrared spectra, shown in Supplementary Fig. 15, and thus resulting in a decrease of Fe(II) content in the nanoparticles. Such a decrease also translated into a loss of absorption in the NIR, partially due to intervalence charge transfer between Fe(II) and Fe(III) (Fig. 4A). Both initial non-oxidized nanoparticles and the oxidized ones were then compared for heating at various concentrations in water upon exposure to the 808 nm laser at 2 W/cm² (Fig. 4B). Magnetite nanoparticles exhibited a higher efficiency at low concentration, with a similar saturation. The degradation of the oxidized nanoparticles (Fig. 4C) was similar to the non-oxidized ones upon global heating, but less marked upon laser (Fig. 2C). The same analysis method was applied to retrieve the local

hot-spot temperature (Fig. 4D), revealing it to be overall lower than for the non-oxidized nanoparticles (Fig. 2F).

Up to now, our data indicates that photothermal irradiation not only triggers nanoparticle degradation through high-temperature hotspots, but also that the associated cell death is enhanced, probably through ferroptosis. To further advance towards this ferroptotic therapeutic concept, we tested the correlation of cytotoxicity and Fe(II) release in a more relevant model using smaller spheroids grown in agarose microwells. Here, the cells were pre-labeled with magnetic nanoparticles prior to spheroid formation to ensure a late endosome localization of the nanoparticles. This smaller spheroid model additionally ensures a complete photothermal irradiation of the cells. The microwells array was made with a 3D-printed stamp with a set of 200-μm micropillars (Fig. 5A). An initial seeding density of about 500 glioblastoma cells (Fig. 5B) matured and self-organized into 150-μm spheroids after 24 h (Fig. 5C).

Spheroid cytotoxicity after laser irradiation was then evaluated using the live/dead cell assay. Control spheroids showed a bright green signal, indicating highly metabolically active cells, with no red dead signal observed (Fig. 5D). On the other hand, the fluorescent response is inversed after 5 min of laser application at a power density of 2.6 W/cm², with all the spheroids being stained only by the red fluorescent dye (Fig. 5E), highlighting high cytotoxicity.

The same procedure was followed for increasing photothermal power density conditions and for both the non-oxidized and oxidized nanoparticles (Supplementary Fig. 16). The quantification of the fluorescent intensity for each of these conditions is shown in Fig. 5F (live signal) and Fig. 5G (dead signal). The laser-induced cytotoxicity of non-oxidized nanoparticles was higher than for the oxidized ones, and was dependent on laser power density. Lastly, we quantified the presence of $Fe^{2+}$ under the same experimental irradiation conditions using the FerroOrange live cell fluorescent dye to detect free $Fe^{2+}$ ions. The dye is not sensitive to the presence of $Fe^{3+}$, other bivalent metal ions, or chelated Fe stored in ferritin, and thus allows us to observe any possible $Fe^{2+}$ nanoparticle degradation products released inside the cells. Figure 5H shows typical fluorescent images of spheroids loaded with non-oxidized and oxidized nanoparticles after photothermal irradiation and subsequent FerroOrange staining. Additional fluorescent images are provided in Supplementary Fig. 17. The FerroOrange fluorescent quantification of all the experimental conditions is shown in Fig. 5I. A clear increase in $Fe^{2+}$ signal can be observed as the irradiation power density increases, a response that is more marked for non-oxidized nanoparticles.

Such an important release of $Fe^{2+}$ could induce Fenton reaction in cancer cells and produce cytotoxic reactive oxygen species (ROS). To assess a possible intracellular ROS generation, spheroids were laser-irradiated and ROS were detected using the green fluorescent marker DCFH-DA. Typical images of the spheroids are shown in Supplementary Fig. 18A–D, indicating an important generation of ROS for the 1.6 W/cm² condition, that decreased for 2.1 W/cm² and was almost zero for 2.6 W/cm². Quantification of the relative fluorescent signal at the single-spheroid level (*n* > 100), shown in Supplementary Fig. 18E, confirms the massive ROS generation at low laser power, decreasing as

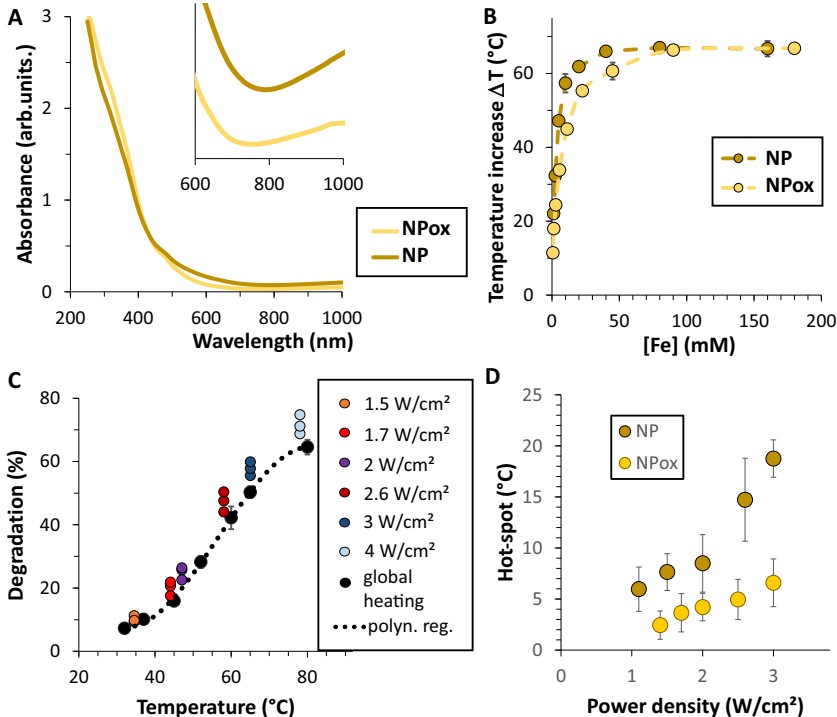

**Fig. 4 | Laser heating and degradation with oxidized nanoparticles (NPox).**
**A** UV–Vis spectra of [Fe] = 1 mM aqueous solutions of non-oxidized (NP) and oxidized (NPox) nanoparticles (NIR close-up behavior shown in inset). **B** Laser heating (temperature increase) for both nanoparticle formulations at a power density of 2 W/cm², and as a function of iron concentration (n = 3). **C** Degradation of oxidized nanoparticles as a function of temperature, either reached by global heating (black curve) or upon laser after 10 min (n = 3). The equivalent curve for non-oxidized nanoparticles is shown in Fig. 2C. **D** Hot-spot temperature (n = 3) generated in the vicinity of oxidized nanoparticles, compared to the ones found for non-oxidized nanoparticles (Fig. 2G). n values represent the number of independent replicas, where data are presented as mean ± SD.

laser power increases. It shows that the released $Fe^{2+}$ can indeed trigger ROS generation through the Fenton reaction, but only if the photothermal effect is small enough to not kill the cells. By contrast, if photothermal irradiation is sufficient to lead to complete cancer cell death, as for instance at 2.6 W/cm² (Fig. 5G), ROS generation is drastically reduced.

Including the Fenton reaction with iron as a catalyst, the ferroptosis pathway is part of a complex cascade of intercellular reactions, with one reliable and well-studied read-out being lipid peroxidation[75]. We thus evaluated the production of lipid peroxides in laser-irradiated spheroids using the specific green fluorescent dye LiperFluo. Typical fluorescent images are shown for different laser power densities in Supplementary Fig. 19A–D, evidencing that the lipid peroxidase presence is significantly higher for the lower power density condition of 1.6 W/cm², whereas it is almost non-existent for the higher power density of 2.6 W/cm². Yet, lipid peroxidation was still more important after treatment with the ferroptosis-inducing agent Erastin (Supplementary Fig. 19E). The visual results were confirmed by quantitative analysis at the single-spheroid level (Supplementary Fig. 19G), overall suggesting the occurrence of ferroptosis for the low laser power density condition, akin to the ROS generation. Similarly, the ferrotptotic oxidative damage mechanism appears to be nullified at high power densities, where it is likely that the high local heating proves to be more efficient to induce cell death. It should be mentioned that lipid peroxidation can be elicited by two different pathways, a non-enzymatic and enzymatic one, with the former being regulated by iron and the latter also with iron playing an important role[76,77]. In addition, lipid peroxidation can result from chemical and physical cell injuries. We do not expect the heating to result in lipid peroxidation, as we observed the highest rate of lipid peroxidation at lower laser power density (Supplementary Fig. 19), while increasing the laser power

density resulted in a lower rate of lipid peroxidation, suggesting that higher temperature conditions, such as those in therapeutic settings, do not induce lipid peroxidation. In addition, the occurrence of lipid peroxidation was assessed following photothermal irradiation of oxidized nanoparticles. It reveals a significantly reduced level of lipid peroxidation (Supplementary Fig. 20), which aligns with the minimal amounts of $Fe^{2+}$ released (Fig. 5I). Finally, the quantification by quantitative PCR of the expression of the two genes SLC7A11 and CHAC1 involved in ferroptosis confirmed this pathway (Supplementary Fig. 21). What appears to be happening is that the photothermal irradiation induces a decrease in intracellular glutathione by reducing its import (downregulation of SLC7A11) and promoting glutathione degradation (upregulation of CHAC1), thus promoting ferroptosis. Importantly, this degradation of glutathione is also observed with Erastin treatment, which correlates with previous studies showing that CHAC1 is one of the genes predominantly overexpressed in Erastin-induced ferroptosis[78].

Taken together, these results confirm that photothermal irradiation of non-oxidized magnetite nanoparticles can trigger the release of $Fe^{2+}$ and the Fenton reaction that leads to ROS generation, further culminating in a synergistic ferroptotic response with lipid peroxidation alongside the temperature increase due to laser exposure.

Such a bimodal laser-mediated thermo-ferroptotic treatment certainly faces the same hurdles than photothermal therapy alone, with the most concerning one being the limited light body penetration. Even if NIR wavelengths have a higher penetration depth over visible wavelengths, external NIR laser application can only penetrate a few mm into deliver sufficient light for treatment. To envisage the thermo-ferroptotic cancer treatment in vivo, two options can be thought off. First, endoscopy can still be used to deliver light deeper, but only to tumors close to the lumen of the vessel. Second, while most

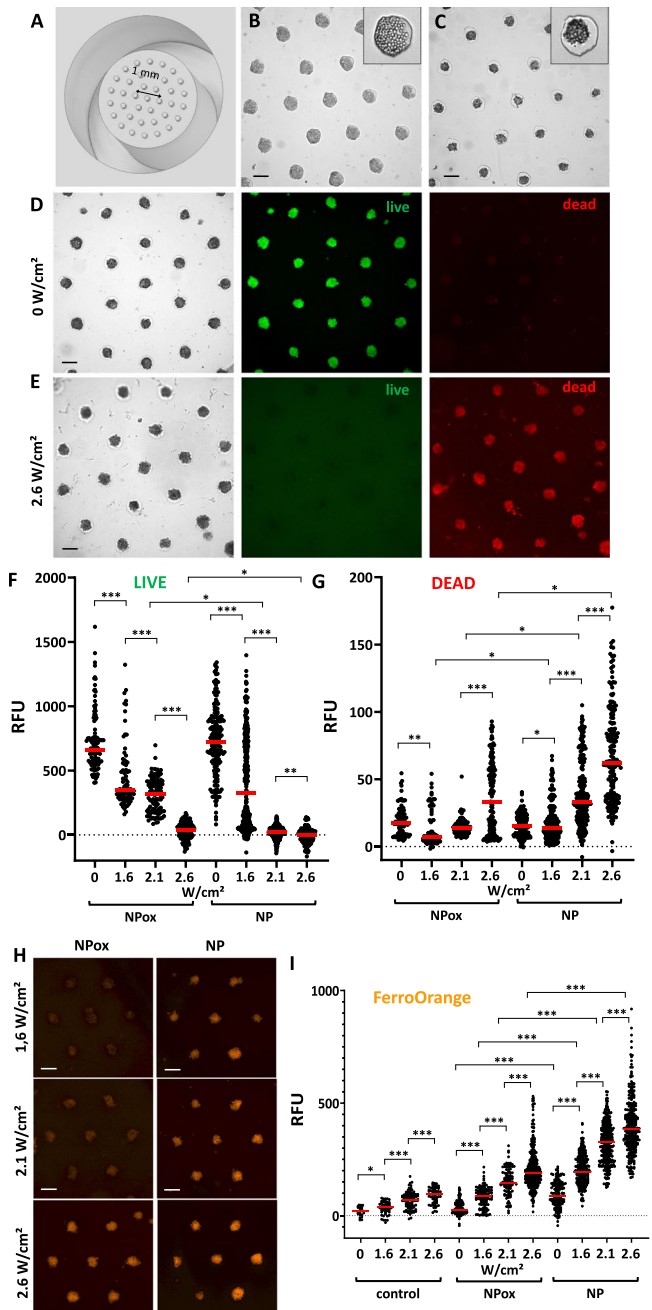

**Fig. 5 | High-throughput analysis of laser-induced degradation ferroptosis.**
**A** 3D-printed stamp with micropillars used to pattern agarose wells inside a 96-well plate for spheroid formation. **B** U87 glioblastoma cells seeded in the microwells, starting from 20,000 cells per well, or equivalent to 540 cells per microwell. **C** Spheroids self-assembling 24 h later. **D, E** Typical live/dead imaging of glioblastoma spheroids without treatment (control condition, **D**) or after laser treatment (5 min−2.6 W/cm², **E**). Left image: bright field; middle: live green fluorescent signal of calcein; right: dead red fluorescent signal of propidium iodide. **F, G** Live green (**F**) and dead red (**G**) fluorescent signal quantification in spheroids after photothermal irradiation at laser power densities of 1.6, 2.1, and 2.6 W/cm², for cells initially labeled with non-oxidized (NP) and oxidized (NPox) nanoparticles ($n = 200$). **H** FerroOrange $Fe^{2+}$ fluorescent imaging of spheroids labeled with non-oxidized (NP) or oxidized (NPox) nanoparticles, after 5 min photothermal irradiation at increasing power densities of 1.6, 2.1, and 2.6 W/cm². **I** FerroOrange quantification for the same conditions ($n = 200$). $n$ values represent the number of analyzed spheroids per condition. Unpaired two-tailed Student's $t$ test was used to evaluate statistical significance, where $*P < 0.05$, $**P < 0.01$, and $***P < 0.001$. Scale bars = 200 μm.

photothermal therapy approaches (including this one) use laser in the first NIR windows (NIR-I, 700–900 nm), the second NIR window (NIR-II, 1000–1700 nm) might be considered for its even deeper transparency compared to NIR-I, associated with higher permissible exposure. Moreover, considering that iron oxide nanoparticles were shown to be efficient in the NIR-II[79], this window might thus be a better option to trigger laser-mediated ferroptosis in vivo. In addition, a major advantage of the dual thermo-ferroptotic treatment can be noted: the efficacy of ferroptosis is maximal at low laser power densities, whereas photothermal therapy is more efficient at higher nanoparticle concentrations, which are typically hard to reach if they are administered intravenously. As a matter of fact, this reflects the local heating at the nanoscale that we report in this work, which is not dependent on the nanoparticle concentration, and that triggers degradation and cytotoxic $Fe^{2+}$ release. Thus, a remotely-triggered ferroptosis effect can prove advantageous over photothermal therapy under in vivo constraints, where irradiation is limited in its penetration, and nanoparticles accumulation can be low.

In summary, the pH and the ratio between nanoparticles and iron chelating agents both mediate nanoparticle degradation, with temperature acting as an accelerator. Moreover, the first proof of concept that laser-induced photothermia can remotely trigger nanoparticle degradation is evidenced. Interestingly, in an ex cellulo model, this condition resulted in a higher degradation compared to a controlled global heating setting, highlighting a higher local heating at the nanoparticle level. This local hot-spot temperature was directly estimated by comparing the degradation rates between the two heating methods. This simple and original way to estimate in operando the elusive hotspot temperature contrasts the complex approaches implemented in the nanoparticles-mediated thermal therapy community.

The exact same laser-mediated nanoparticle degradation was then validated in the cellular environment of 3D glioblastoma tumoroids, with differences in efficiency between the localization of the nanoparticles (early or late endosomal), in line with the changes in pH of the reached compartments. Lastly, using a high-throughput spheroid model, the potential of this degradation was demonstrated through a combined photothermal-ferroptosis therapy with iron oxide nanoparticles by showing the laser power-dependent systematic release of cytotoxic Fe(II). Overall, the remotely-triggered local nanoparticle temperature increase stands as a decisive factor in its degradation state, with nanoparticle core composition being a critical factor in the cellular ferroptotic response. To the best of our knowledge, the work presented here offers the first insight into the quantification of the release of cytotoxic $Fe^{2+}$ from iron oxide nanoparticles upon laser exposure, as well the first determination of the local temperature at the nanoparticle level, obtained from the magnetic signature of the nanoparticles upon degradation.

## Methods

### Nanoparticle synthesis

Magnetic nanoparticles were synthesized via microwave-assisted synthesis. Briefly, 400 mg of iron (III) acetylacetonate (>99.9%) (Sigma Aldrich) were dissolved in 10 mL of benzyl alcohol anhydrous (99.8%) (Sigma Aldrich) within a 30 mL monowave glass vial and placed in a monowave 300 (Anton Paar). The temperature of the suspension was increased up to 250 °C in 20 min and then maintained constant for 30 min. The resulting suspension was precipitated using a neodymium magnet and the precipitate was washed successively with dichloromethane, sodium hydroxide (1 M), ethanol and pH = 7 water (three times). A last washing step was performed in acidic water (pH 2) and the nanoparticles were separated by ultracentrifugation using Amicon® ultra centrifugal filters (30 kD) at 2200 × g during 10 min. Finally, the magnetic nanoparticles were resuspended in acidic water (pH = 2). Iron oxide nanoparticle majorly formed as magnetite ($Fe_3O_4$)

were obtained. To oxidize them into maghemite-like ($\gamma$-$Fe_2O_3$) nanoparticles, a fraction of the previous batch was heated at 80 °C under reflux under magnetic stirring for 16 h. Fourier Transform Infrared (FTIR, Nicolet 380, Thermo Fisher Scientific) spectroscopy coupled with the OMNIC 8.1.210 software, and a vibrating-sample magnetometer (VSM, Quantum Design Versalab) were used to determine the oxidation state of the iron nanoparticle solution. For nanoparticles coating, citric acid (Sigma Aldrich) was used in excess with a mass of citrate five times higher than the mass of nanoparticles and was dissolved in water at pH = 2. The coating molecule solution and nanoparticle dispersion were mixed and left to agitate for two hours under stirring. The pH was then adjusted to 7 with sodium hydroxide at 1 M and left for two hours to equilibrate. Finally, nanoparticles were magnetically sorted three times in water (pH = 2), using a neodymium magnetic disc, then ultra-centrifuged with deionized water at pH = 7 in Amicon® ultra centrifugal filters (30 kD) during 10 min at $2200 \times g$.

### Degradation in aqueous dispersions
Solutions roughly mimicking the lysosomal environment were prepared by mixing nanoparticles in a buffer with acidic conditions (with pH ranging from 4 to 6.5) and an iron chelating agent (citrate, from 1 to 20 mM) to be at a final concentration of [Fe] = 2 mM of iron oxide nanoparticles.

### Magnetization measurement
Magnetization was measured by a magnetic sensor developed by Magnisens SA (MIAtek) for diagnostics tests. The analysis is based on the non-linear superparamagnetic magnetization of the nanoparticles. Briefly, an alternating magnetic field is applied to the sample at two different frequencies: $f_1 = 100$ kHz and $f_2 = 100$ Hz, at amplitudes of 1 and 20 mT for both. The concept is based on the magnetization of the nanoparticles with the low-frequency field, and then switch this magnetization sinusoidally at high frequency so that the sensing coil voltage is modulated by both frequencies. Combinatorial Fourier transform analysis then provides the third derivative of the sample magnetization around the zero magnetic field and at room temperature, with a detection threshold within 10 µemu. The measurement lasts less than 30 s, on a volume of 100 µL, which can be done either in real time or sampled as many times as necessary.

### Laser-induced thermometric measurements
Heating profiles of aqueous solutions were obtained by placing suspensions of 50 µL of magnetic nanoparticles diluted in lysosomal-like solution at [Fe] = 2 mM in a 0.5 mL Eppendorf tube, at a working distance of 4.5 cm between the laser source line and the liquid surface. Samples were irradiated with an 808-nm laser (Laser Components) at power densities between 1 to 3.2 W/cm² for 10 to 30 min. The increase in temperature was measured in real time using an infrared thermal camera (FLIR A615) and processed with the FLIR ResearchIR software.

### Cell culture
U87 MG human glioblastoma cells (89081402, Sigma Aldrich) were cultured in Dulbecco's Modified Eagle Medium (DMEM, Gibco) supplemented with 10% fetal bovine serum and 1% penicillin–streptomycin (Thermo Fisher Scientific) at 37 °C in a humidified incubator with 5% $CO_2$. Upon reaching 80% confluence, cells were detached using 0.05% trypsin-EDTA (Gibco) prior to use for spheroids formation.

### Large spheroids formation and nanoparticle labeling
In total, 400,000 cells were dispersed in 1 mL of culture medium in a 15 mL Falcon tube and centrifuged at $230 \times g$ for 5 min to form a pellet. The pellets were left in an incubator for two days to mature and form cohesive spheroids. A labeling solution was prepared with nanoparticles diluted at [Fe] = 2 mM in serum-free RPMI culture medium (Thermo Fisher Scientific) supplemented with 5 mM free citrate (to avoid nanoparticle precipitation). Spheroids at day two of maturation were placed in this solution for 30 min at 4 °C under slow stirring to allow nanoparticles to interact with the entire spheroid surface. The spheroids were then rinsed thoroughly in serum-free RPMI medium and incubated for different durations (30 min, 2 h, and 4 h) in complete DMEM medium at 37 °C before further processing to allow nanoparticles to be internalized at different stages of the endocytosis process (from surface/early endosomes to endosomes/late endosomes).

### Prussian Blue staining
Prussian Blue staining, consisting of 5% potassium ferrocyanide (Sigma Aldrich) in 10% hydrochloric acid (Sigma Aldrich), was used to observe the localization of iron in cells. Spheroids were fixed with 4% paraformaldehyde and then included in 1.5% agarose in deionized water. Spheroids were then cut in slices of 7 µm directly from the agarose mold (HistIM platform, Cochin Institute, France) and were stained with Prussian Blue for 35 min and counterstained with nuclear fast red (Sigma Aldrich) for 15 min. Images were taken using an Olympus B optical microscope.

### Transmission electron microscopy imaging
Spheroids were analyzed by TEM to observe the exact nanoparticles localization within the cells at the nanoscale level. The spheroids were washed with cacodylate buffer (0.2 M), then fixed with 5% glutaraldehyde in 0.1 M cacodylate buffer for 30 min at room temperature, and washed once again in the same buffer. Samples were then contrasted with Oolong Tea Extract at 0.5% diluted in cacodylate buffer, post-fixed with 1% osmium tetroxide containing 1.5% potassium cyanoferrate and then dehydrated in graded ethanol baths. Samples were then included in Epon epoxy resin, ultra-sectioned (70 nm) and deposited onto copper grids for observation with a HITACHI HT 7700 electron microscope operating at 120 kV (INRA, Plateau MET, Jouy-en-Josas, France).

### Iron quantification by inductively coupled plasma atomic emission spectroscopy
Total iron content per spheroid was determined by inductively coupled plasma atomic emission spectroscopy (Spectrogreen, SPECTRO, Germany). Spheroids were digested in 290 µL of 69% nitric acid (Sigma, trace metal basic grade) for 2 days. To quantify nanoparticle internalization in individual cells (before spheroids formation in microwells), cell number was counted using a Malassez chamber, and samples were also digested in 290 µL of 69% nitric acid. All solutions were then diluted in filtered ultrapure water to obtain a final 2% nitric acid solution ready for analysis.

### Photothermal irradiation of large spheroids and live/dead fluorescent staining and imaging
For photothermal experiments with the large spheroids model, eight spheroids were placed at the bottom of a 2 mL Eppendorf tube with 100 µL of culture medium. Spheroids were irradiated at a power density of 2 W/cm², taking care that all spheroids were on the optical path of the laser, with none above one another. The LIVE/DEAD™ Cell Imaging Kit (R37601, Invitrogen™) was used to assess cell viability in live spheroids. Spheroids at day two of maturation were stained with Live Green/Dead Red staining solution mixed at a 1:2 dilution in complete culture medium in non-adherent dishes for two hours in an incubator at 37 °C, according to the vendor's instructions. The spheroids were then imaged using a Nikon Eclipse microscope (Nikon®) coupled with fluorescence and a CoolSnap HQ2 camera (Photometrics), with a 10× and 20× objective. Images were processed using the ImageJ software.

### Spheroid formation using agarose microwells, laser irradiation, live/dead, and FerroOrange stainings

U87 glioblastoma cells were incubated with [Fe] = 2 mM of either non-oxidized magnetite nanoparticles or oxidized maghemite ones in DMEM cell medium and left overnight in a 37 °C incubator. The following day, cells were detached using 0.05% trypsin-EDTA and counted. A 3D-printed stamp with micropillars of 200 μm diameter and 200 μm in height was used to print molds into the wells of a 96-well plate containing 50 μL of 2% agarose (A0576, Sigma Aldrich) in phosphate buffered saline. The agarose was then left to solidify for 5 min and the stamp was removed, leaving an array of molds (microwells) in the wells. The 96-well plate was then sterilized in UV light for 30 min before cell seeding (20,000 cells per well, corresponding to approximately 500 cells per microwell) and centrifugation. Cells were allowed to mature for 24 h before photothermal irradiation. Irradiation was performed directly within the 96-well plate containing the microwells, with the laser placed at 3 cm above each well using a custom-made laser holder (Supplementary Fig. 22), irradiating the well with power densities of 1.6, 2.1, and 2.6 W/cm². Live/dead staining was performed as previously indicated, with 100 μL per well, and imaging was performed on an EnSight® Multimode Microplate Reader (Perkin Elmer) coupled with the Kaleido 3.0 software for image acquisition, and the ImageJ software was used for processing the acquired images and for fluorescence quantification. Iron (II) fluorescent imaging and quantification was performed after laser irradiation with the FerroOrange fluorescent probe (Millipore BioTracker™ FerroOrange Live Cell Dye, # SCT210), following the specified protocol by the vendor. Before staining, the spheroids were washed thoroughly with washing buffer to remove traces of extracellular iron. The spheroids were then incubated at 37 °C for 30 min in a 1 μM FerroOrange solution diluted in Hank's Balanced Salt Solution (Thermo Fisher Scientific). After the incubation, the cells were rinsed twice with washing buffer and placed in observation buffer before imaging with an EnSight® Multimode Microplate Reader. Images were analyzed using the ImageJ software.

### Statistical analysis

Data are presented as the mean of replicates, with plots showing mean ± standard deviation. Significance between two groups was determined using an unpaired two-tailed Student's $t$ test. For all values, a minimum of 95% confidence level was considered significant, with *$P < 0.05$, **$P < 0.01$ and ***$P < 0.001$. The number of independent experiments was systematically superior to three.

### Reporting summary

Further information on research design is available in the Nature Portfolio Reporting Summary linked to this article.

## Data availability

The full image dataset is available from the corresponding author upon request. Source data are provided with this paper.

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

## Acknowledgements
This work was funded by the European Union (ERC-2019-CoG project NanoBioMade 865629 and ERC-2022-POC1 project BioMag 101069301). It has also received the support of "Institut Pierre-Gilles de Gennes" (laboratoire d'excellence, Equipex, "Investissements d'avenir" program ANR-10-IDEX-0001-02 PSL and ANR-10-LABX-31-34). The authors thank Giacomo Gropplero for help in 3D printing and acknowledge the CNanoMat platform of Sorbonne Université Paris Nord for UV–vis and FTIR spectroscopies, the HistIM platform at Cochin Institute for histology analysis (Maryline Favier), the IPGP multidisciplinary program PARI and Paris–IdF region SESAME Grant no. 12015908 for ICP–AES analyses (Laure Cordier) and the MIMA-2 platform from INRAE, Jouy-en Josas, for TEM imaging (Christine Péchoux).

## Author contributions
C.W. and Y.L. conceptualized the study. All authors performed experiments. C.W. and J.E.P. wrote the manuscript.

## Competing interests
The authors declare no competing interests.
