## [Peer Review File · Nature Communications]

Reviewers' Comments:

Reviewer #1:

Remarks to the Author:

Reviewer Recommendation Term: Reject

The manuscript titled "Photothermia at the nanoscale induces ferroptosis via nanoparticle degradation" proposed a novel method to demonstrate that laser-induced photothermal effect can generate local hot-spot which trigger the degradation of iron oxide nanoparticles. Such a treatment lead to higher nanoparticle degradation compared to global heating with the same temperature. In addition, using a tumor spheroid model, the authors analyzed the degradation of nanoparticles and the release of cytotoxic Fe(II), as well as the antitumor capability of the proposed photothermal-ferroptosis combinational approach.

This research article shows potential significance on the research and application of nanoparticles-based hyperthermia cancer therapy. However, considering the complexity of the biological system and nano-bio interactions, much stronger and sufficient evidences are required to support the hypotheses and conclusions in this manuscript. Therefore, the manuscript in current version cannot meet the level for publishing in Nature Communications.

To further improve the quality of this manuscript, some questions and suggestions are listed as follows:

1. Figure 3, In addition to incubation duration and temperature, other important factors may also affect the cellular/tumoral internalization and localization of nanoparticles, such as the phenotypic and functional heterogeneity of cancer cells, the location of target cells in tumor spheroids, etc. The authors are suggested to use more comprehensive methods for cellular analysis, such as statistical methods, to demonstrate the reliability of the "spheroid endocytosis model" in this manuscript.
2. The authors stated in Section 2.4 that "As thermal equilibrium is supposed to be reached in the immediate vicinity of the nanoparticles, it is unlikely that the increase of cell death is due to the higher local heating, and is more likely to be the result of the higher degradation that could induce a ferroptotic response with release of Fe(II)." In addition to the photothermal effect, iron oxide nanoparticles could also induce Fenton reaction in cancer cells and produce cytotoxic reactive oxygen species, of which the reaction efficiency is closely related to temperature and pH. In this manuscript, the interactions between the photothermal and Fenton effect, as well as their contributions to tumor cell death should be clarified and elaborated more.
3. Cell death mechanism studies involving ferroptotic pathways were missed, although "ferroptosis" seemed to be one of the key words in this manuscript.

Reviewer #2:

Remarks to the Author:

The main objective of the work is the local temperature determination at the nanoscale by measuring the degradation of the particles after photothermal application. They also demonstrate that it is possible to trigger nanoparticle degradation after a laser-induced hot-spot temperature in vitro and this effect could be used as anti-cancer therapy through iron release and ferroptosis. Same authors already analysed this effect but not in a quantitative way. Magnetic (hyper) thermia or photothermia? Progressive comparison of iron oxide and gold nanoparticles heating in water, in cells, and in vivo, 2018/9, Advanced Functional Materials

Other authors have shown the degradation in lisosomal-like suspensions of iron oxide nanoparticles and in the presence of magnetic and photothermal treatments [48, 53].

Here, authors use that degradation to determine the temperature It is indeed a simple and original way of measuring temperature at the surface of the nanoparticles.

Between the attempts made using molecular thermometers, authors forgot the one analysing the enzymatic activity of fluorescent proteins oriented attached to the magnetic nanoparticles, although the reference is included in other general sentence [44].

Some minor remarks:

- Authors should comment on the effect of the coating, effect of the protein corona, aggregation state that can also affect the heating properties of the colloid.

- Non-oxidized nanoparticles and the oxidized ones should be characterized by IR spectra or Raman that can be used to monitor the oxidation degree (250-850 cm⁻¹). Also, by Mossbauer [J. Phys. D: Appl. Phys. 2017, 50, 265005, DOI: 10.1088/1361-6463/aa73fa]. Being Magnetite/maghemite a solid solution that may have different degrees of oxidation depending on the particle size and the coating, authors should provide the characterization of their samples.
- Limitations of the photothermal therapy should be indicated such as the light limited body penetration.
- Authors claim that: "Standard magnetometry methods typically require large-scale equipment with high measurement times. Additionally, as samples require fixation, the analysis of time-dependent properties is not possible". VSM is a simple and relatively inexpensive equipment for the fields required in this study and at room temperature. It is reliable providing information about the magnetic moment of a material as a function of temperature, field, and crystal orientation. This analysis is done via a small-amplitude alternating magnetic field (H) applied to colloidal suspensions of nanoparticles. In addition, low-field magnetic susceptibility can be measured using a Bartington MS 2B dual-frequency sensor (Bartington Instruments Ltd., Oxford, UK), for example.

Reviewer #3:

Remarks to the Author:

This paper titled "Photothermia at the nanoscale induces ferroptosis via nanoparticle degradation" is reporting that iron oxide nanoparticles can have a degradation with laser heating photothermia. During the study, global heating and local heating were defined for obtaining hot spot temperatures. Those correlation process logically performed and it is showing a meaningful results. Since there is a challenge to measure the hot spot temperature during thermoseed heating, those results and method are very interesting. Degradation dependent photothermia heating behavior of iron oxide nanoparticles can be observed in the different stage of nanoparticles cellular uptake too. Eventually, authors hypothesized that the photothermia triggered degradation will release Fe²⁺ to induce ferroptosis cell killing. Overall, the topic is interesting and research design/data of laser heating and degradation are well-presented. However, the main concern is the application part. The ferroptosis will be strongly associated with Fe²⁺ ion but the cell death mechanism is complex. Rigorous data that can prove the ferroptosis induction from the photothermia with nanoparticles is strongly required. Additionally, some details are missing in the manuscript.

- 1) The developed magnetic sensor is not well described. Additional references, equation, and validation data will be helpful.
- 2) It is not clear the morphological change of nanoparticles after the photothermia degradation.
- 3) Is there any possibility of crystallinity change of iron oxide nanoparticles after photothermia? X-ray diffraction data will be helpful.
- 4) Need more discussion on only heating effect on Fe²⁺ intracellular ions.
- 5) Additional ferroptosis markers should be tested.
- 6) More discussion on in vivo application of the photothermia induced ferroptosis induction is essential.
- 7) There are some papers related with photo-initiated ferroptosis, caused by released ferrous ions from iron oxide. Authors may add more information that can represent the novelty of this study compared to the published reports.

Response to Reviewer #1

The manuscript titled "Photothermia at the nanoscale induces ferroptosis via nanoparticle degradation" proposed a novel method to demonstrate that laser-induced photothermal effect can generate local hot-spot which trigger the degradation of iron oxide nanoparticles. Such a treatment lead to higher nanoparticle degradation compared to global heating with the same temperature. In addition, using a tumor spheroid model, the authors analyzed the degradation of nanoparticles and the release of cytotoxic Fe(II), as well as the antitumor capability of the proposed photothermal-ferroptosis combinational approach.

This research article shows potential significance on the research and application of nanoparticles-based hyperthermia cancer therapy. However, considering the complexity of the biological system and nano-bio interactions, much stronger and sufficient evidences are required to support the hypotheses and conclusions in this manuscript. Therefore, the manuscript in current version cannot meet the level for publishing in Nature Communications.

We thank the reviewer for positive assessment of our work significance, and we acknowledge the concern for more evidence of the mechanism claimed. We have now performed additional experiments that support ferroptosis, as introduced in details in the answer to comments 2) and 3);

To further improve the quality of this manuscript, some questions and suggestions are listed as follows:

1. Figure 3, In addition to incubation duration and temperature, other important factors may also affect the cellular/tumoral internalization and localization of nanoparticles, such as the phenotypic and functional heterogeneity of cancer cells, the location of target cells in tumor spheroids, etc. The authors are suggested to use more comprehensive methods for cellular analysis, such as statistical methods, to demonstrate the reliability of the "spheroid endocytosis model" in this manuscript.

We thank the reviewer for this insight. Regarding the location of the target cells in our spheroid model, we indeed expect the nanoparticles to penetrate mostly into the proliferating cells located in the outer layer of the spheroid, as shown in the Prussian Blue staining in Figure 3C. Further, we are aware the single spheroid model we provide in this article is limited in number, and can hardly provide data in phenotypic variation and cell heterogeneity. However, we believe that within the scope of the data that we present on the laser-mediated degradation of magnetic nanoparticles (Figure 3L, showing statistically significant differences), this model is sufficient to unravel the effect of intracellular localization. By contrast, to integrate the functional heterogeneity of cancer cells, we next moved to a high-throughput model and microfabricated spheroids, and did all further analyses over 100 single spheroids (Figure 5 and Figures S18 and S19). Nevertheless, in order to lend credence to the presented data on intracellular localization, we sought out to evaluate whether cell viability in this spheroid model would follow a similar response to thermal injury depending on the nanoparticles localization, with the results nicely correlating with that of degradation: cell viability was significantly lower when nanoparticles were expected to be located in early endosomes, for both global heating and laser irradiation settings, with cell death being more marked after a longer nanoparticle processing and late endosome localization. These results go in line with our hypothesis, and are now presented in supplementary Figure S14.

Figure S14: Cell viability quantification in U87 tumor spheroid model after global heating and laser irradiation after an incubation of nanoparticles of 30 minutes (early endosomes) or 4 hours (late endosomes). Cell viability was assessed using a live/dead cell assay, and calculated via cell counting and fluorescence imaging. Cell viability was overall higher for controlled global heating compared to photothermal application. A significant difference in cell viability was also apparent between early and late endosomes.

2. The authors stated in Section 2.4 that “As thermal equilibrium is supposed to be reached in the immediate vicinity of the nanoparticles, it is unlikely that the increase of cell death is due to the higher local heating, and is more likely to be the result of the higher degradation that could induce a ferroptotic response with release of Fe(II).” In addition to the photothermal effect, iron oxide nanoparticles could also induce Fenton reaction in cancer cells and produce cytotoxic reactive oxygen species, of which the reaction efficiency is closely related to temperature and pH. In this manuscript, the interactions between the photothermal and Fenton effect, as well as their contributions to tumor cell death should be clarified and elaborated more.

We thank the reviewer for this insightful comment. We agree that this sentence may have been misleading, especially because despite the heat equilibrium at the spheroid level, high local heating is achieved, and PTT definitely contributes to cell death. However, we also totally agree with the reviewer that the nanoparticles could then induce Fenton reaction in cancer cells and produce cytotoxic reactive oxygen species. We have therefore removed the referenced sentence, and have now clarified the dual mechanism concept of cell death consisting of both PTT and the Fenton reaction.

As a matter of fact, this is the mechanism we were assuming was occurring, when we were claiming that ferroptosis took place. Our data presented in Figure 5I showing an increased level of Fe²⁺ were supposed to support the claim of Fenton reaction and ferroptosis, yet we acknowledge that this was probably not particularly clear, with the Fenton reaction not specifically mentioned. We have thus rewritten this part in the revised manuscript which now introduces the concept of Fe²⁺-triggered cytotoxic Fenton reaction. Moreover, to further support further this claim, we have assessed ROS generation in cells using the green fluorescent marker DCFH-DA (Dojindo, R252), detectable using plate reader. In brief, spheroids were formed in microwells (using a 6-well plate stamp to generate 500 spheroids per well) from an initial seeding density of 500 cells labelled with nanoparticles, and

matured over 48 hours. They were then transferred to a 96-well plate (100 per well) and exposed to laser with the same protocol as introduced in the Methods. 1 hour after exposure, the spheroids were washed with HBSS and incubated with the ROS dye for 30 min, washed again and observed with a plate reader (green channel). Remarkably, it clearly demonstrates that the release of Fe^{2+} via nanoparticles degradation can lead to Fenton-like reaction with massive ROS generation, but only for the low laser power density of 1.6 W/cm^2 . For the higher power density of 2.1 W/cm^2 , ROS are still generated, but to a lesser extent, and no ROS could be detected at 2.6 W/cm^2 . While this could be found counterintuitive with the increased release of Fe^{2+} with laser power density, it in fact is quite logical if one considers that a laser power density at 2.1 W/cm^2 and even more at 2.6 W/cm^2 is supposed to produce high local heating and kill cells, as observed with the live dead staining shown in Figure 5G of the submitted manuscript. These new data evidencing ROS generation are now included as new supplementary Figure S18, and a full explanation has been added in the revised manuscript as follows:

“Such an important release of Fe^{2+} could induce Fenton reaction in cancer cells and produce cytotoxic reactive oxygen species (ROS). To assess possible intracellular ROS generation, spheroids were laser irradiated and ROS were detected using the green fluorescent marker DCFH-DA. Typical images of the spheroids are shown in supplementary Figure S18A-D, indicating an important generation of ROS for the 1.6 W/cm^2 condition, that decreased for 2.1 W/cm^2 , and was almost zero for 2.6 W/cm^2 . Quantification of the relative fluorescent signal at single spheroid level ($n > 100$), shown in Figure S18E, confirms the massive ROS generation at low laser power decreasing with laser power. It shows that the released Fe^{2+} can indeed trigger ROS generation through the Fenton reaction, but only if the PTT effect is small enough to not kill the cells. By contrast, if PTT is sufficient to lead to complete cancer cell death, as for instance at 2.6 W/cm^2 (Figure 5G), ROS generation is drastically reduced.”

Figure S18: ROS generation in cells. In brief, spheroids were formed in microwells (using a 6-well plate stamp to generate 500 spheroids per well) from an initial seeding density of 500 cells labelled with nanoparticles, and matured over 48 hours. They were then transferred to a 96-well plate (100 per well) and exposed to laser with the same protocol as introduced in the Methods, at the three different power densities of 1.6, 2.1, and 2.6 W/cm², and for 5 min. 1 hour after exposure, the spheroids were washed with HBSS and incubated with green fluorescence ROS dye (DCFH-DA, Dojindo, R252) for 30 min according to supplier's instructions, washed again and observed with a plate reader (green channel). **A.** Typical images of the spheroids, indicating an important generation of ROS for the 1.6 W/cm² condition, that decreased for 2.1 W/cm², and was almost zero for 2.6 W/cm². **B.** Quantification of the relative fluorescent signal at single spheroid level (n>100) confirms the massive ROS generation at low laser power decreasing with laser power. *** p<0.0001.

3. Cell death mechanism studies involving ferroptotic pathways were missed, although “ferroptosis” seemed to be one of the key words in this manuscript.

We agree with the reviewer that more evidence was needed to strengthen our ferroptosis claim. Besides, even if ferroptosis was pinpointed for the reason that the process is iron-dependent and can be prevented by iron chelators, the mechanisms at play are still controversial. In particular, it is generally admitted that Fenton reaction and ferroptosis are tightly interwoven, but it still seems that they can also take place independently. If iron and lipid peroxides remain the two major participants in ferroptosis, lipid peroxidation is the ultimate result of ferroptosis, while iron is generally described as a catalyst or regulator. We thus also tested lipid peroxidation after laser application, using the fluorescent dye LiperFluo. Remarkably, it led to the similar observation to that of ROS generation: at low laser power density of 1.6 W/cm², an important fluorescent signal is detected, revealing accumulation of lipid peroxides, and thus the occurrence of ferroptosis, while the signal decreases with increased laser power, and was inexistent at 2.6 W/cm² high power. Again, this is probably due to the interplay with PTT, killing the cells more efficiently at high local temperature, precluding any ferroptotic oxidative mechanism to happen.

Images of lipid peroxidation and quantitative analyses of single spheroids relative fluorescence intensity (n>300) are shown in new supplementary Figure S19.

Both are introduced in the revised manuscript as follows:

“Including the Fenton reaction with iron as a catalyst, the ferroptosis pathway is part of a complex cascade of intercellular reactions, where one reliable and well-studied read-out being lipid peroxidation.⁷⁵ We thus evaluated the production of lipid peroxides in laser-irradiated spheroids using the specific green fluorescent dye LiperFluo. Typical fluorescent images are shown for different laser power densities in supplementary Figure S19A-D, evidencing that the lipid peroxidase presence is significantly higher for the lower power density condition of 1.6 W/cm², whereas it is almost non-existent for the higher power density of 2.6 W/cm². Yet, lipid peroxidation was still more important after treatment with the ferroptosis-inducing agent Erastin (Figure S19E). The visual results were confirmed by quantitative analysis at the single-spheroid level (Figure S19G), overall suggesting the occurrence of ferroptosis for the low laser power density condition, akin to the ROS generation. Similarly, the ferroptotic oxidative damage mechanism appears to be nullified at high power densities, where it is likely that the high local heating proves to be more efficient to induce cell death.”

Figure S19: Similarly to the previous ROS analysis setup, spheroids were irradiated at increasing laser power densities of 1.6, 2.1 and 2.6 W/cm² for a duration of 5 minutes. 3 hours later, the spheroids were incubated for 30 minutes with the fluorescent dye LiperFluo (Dojindo) according to supplier's instructions. Erastin (Sigma-Aldrich, E7781) was used as positive control to induce ferroptosis. Spheroids were then treated with Erastin (5μM) for 48 hours before incubation with LiperFluo. **A.** Typical fluorescent images of control, and the three laser conditions, and the Erastin condition. Lipid peroxidase presence is high for the lower power density condition (yet lower than with Erastin administration), whereas it is almost non-existent for the higher power density. **B.** Quantitative analyses of single spheroids relative fluorescence intensity (n>300). *** p<0.0001.

Response to Reviewer #2

The main objective of the work is the local temperature determination at the nanoscale by measuring the degradation of the particles after photothermal application. They also demonstrate that it is possible to trigger nanoparticle degradation after a laser-induced hot-spot temperature in vitro and this effect could be used as anti-cancer therapy through iron release and ferroptosis. Same authors already analysed this effect but not in a quantitative way. Magnetic (hyper) thermia or photothermia? Progressive comparison of iron oxide and gold nanoparticles heating in water, in cells, and in vivo, 2018/9, Advanced Functional Materials

Other authors have shown the degradation in lysosomal-like suspensions of iron oxide nanoparticles and in the presence of magnetic and photothermal treatments [48, 53].

Here, authors use that degradation to determine the temperature. It is indeed a simple and original way of measuring temperature at the surface of the nanoparticles.

We thank the reviewer for the positive evaluation of our work. Indeed, we believe that besides the possibility to laser-trigger degradation remotely, the main originality of this work is the local temperature determination in the vicinity of the nanoparticles.

Between the attempts made using molecular thermometers, authors forgot the one analysing the enzymatic activity of fluorescent proteins oriented attached to the magnetic nanoparticles, although the reference is included in other general sentence [44].

We thank the reviewer for the remark, and we have now cited this reference 44 among the attempts made using molecular thermometers.

Some minor remarks:

- Authors should comment on the effect of the coating, effect of the protein corona, aggregation state that can also affect the heating properties of the colloid.

We thank the reviewer for the suggestion and have briefly introduced these effects in the revised manuscript

”Notably, iron oxide nanoparticles are potent agents for both modalities, with photothermal conversion being the most efficient thermal treatment at low doses of nanoparticles in the cellular environment. Contrary to plasmonic nanoparticles, their photo-heating capacity does not significantly depend on aggregation or coating.”

- Non-oxidized nanoparticles and the oxidized ones should be characterized by IR spectra or Raman that can be used to monitor the oxidation degree (250-850 cm^{-1}). Also, by Mossbauer [J. Phys. D: Appl. Phys. 2017, 50, 265005, DOI: 10.1088/1361-6463/aa73fa]. Being Magnetite/maghemite a solid solution that may have different degrees of oxidation depending on the particle size and the coating, authors should provide the characterization of their samples.

We thank the reviewer for the suggestion. We chose to associate FTIR and TEM, to characterize respectively the magnetite/maghemite structures, the coating efficiency and the crystalline size. Figure S3 and Figure S1 (C, D) now show the FTIR spectra and TEM images for both nanoparticles.

Both NP and NPox exhibit the characteristic Fe-O vibration band within the 800-450 cm^{-1} region. For the NP sample, the spectrum is characteristic of magnetite (Fe_3O_4) with a single peak at 580 cm^{-1} while

the oxidized sample NPox exhibit a peak at 630 cm^{-1} which is the maghemite signature, with the formation of a spinel phase.

Besides, by analyzing the area of the characteristic bands at 1630 and 580 cm^{-1} of respectively the C=O and Fe-O regions, we find a constant ratio area A_{1630}/A_{580} between NP and NPox samples. This indicates an equivalent amount of citrate for both.

Figure S3: Fourier Transform Infrared (FTIR, Tensor 27 Bruker) of citrate coated non oxidized (NP) (blue curve) and oxidized (NPox) (red curve) nanoparticles. The FTIR spectra were recorded on a using transmission mode trough KBr pellets: Nanoparticles solution is mixed to KBr at a ratio of 5 mg FeOx for 100 mg of KBr and freeze dried overnight, and the KBr pellet is next obtained under pressure. Both NP and NPox exhibit the characteristic Fe-O vibration band within the $800\text{-}450\text{ cm}^{-1}$ region. The NP spectrum is similar to magnetite with a single peak at 580 cm^{-1} while the oxidized sample NPox exhibit a peak at 630 cm^{-1} , signature of maghemite with the formation of a spinel phase.

We have also included in previous Figure S1 containing TEM images and size distribution of the non-oxidized nanoparticles the same analysis with oxidized ones, revealing the same size distribution after oxidation.

Figure S1: Transmission Electron microscopy imaging of citrate coated iron oxide nanoparticles (A) and associated size distribution (B) and same analysis for the oxidized nanoparticles (C,D); Mean diameters for both nanoparticles are similar, 9.6 ± 2.1 nm and 9.4 ± 2 nm for non-oxidized and oxidized nanoparticles, respectively.

- Limitations of the photothermal therapy should be indicated such as the light limited body penetration.

This has now been indicated in the revised manuscript, together with a discussion on the limitation to apply such protocol *in vivo*.

“Such a bimodal laser-mediated thermo-ferroptotic treatment certainly faces the same hurdles than photothermal therapy alone, with the most concerning one being the limited light body penetration. Even if near-infrared (NIR) wavelengths have a higher penetration depth over visible wavelengths, external NIR laser application can only penetrate a few mm in to deliver sufficient light for treatment. To envisage the thermo-ferroptotic cancer treatment *in vivo*, two options can be thought off. First, endoscopy can still be used to deliver light deeper, but only to tumors close to the lumen of the vessel. Second, while most PTT works (including this one) use laser in the first NIR windows (NIR-I, 700–900 nm), the second NIR window (NIR-II, 1000–1700 nm) might be considered for its even deeper transparency compared to NIR-I, associated with higher permissible exposure. More the less, iron oxide nanoparticles were shown to be efficient in the NIR-II.⁷⁶ NIR-II might thus be a better option to trigger laser-mediated ferroptosis *in vivo*. In addition, a major advantage of the dual thermo-ferroptotic treatment can be noted: the efficacy of ferroptosis is maximal at low laser power densities, whereas photothermal therapy is more efficient at higher nanoparticle concentrations, which are typically hard to reach if they are administered intravenously. As a matter of fact, this reflects the local heating at the nanoscale that we report in this work, which is not dependent on the nanoparticle concentration, and that triggers degradation and cytotoxic Fe^{2+} release. Thus, a remotely-triggered ferroptosis effect can prove advantageous over photothermal therapy under *in*

vivo constraints, where irradiation is limited in its penetration, and nanoparticles accumulation can be low.”

- Authors claim that: “Standard magnetometry methods typically require large-scale equipment with high measurement times. Additionally, as samples require fixation, the analysis of time-dependent properties is not possible”. VSM is a simple and relatively inexpensive equipment for the fields required in this study and at room temperature. It is reliable providing information about the magnetic moment of a material as a function of temperature, field, and crystal orientation. This analysis is done via a small-amplitude alternating magnetic field (H) applied to colloidal suspensions of nanoparticles. In addition, low-field magnetic susceptibility can be measured using a Bartington MS 2B dual-frequency sensor (Bartington Instruments Ltd., Oxford, UK), for example.

We thank the reviewer for the comment. We agree that our sentence could be misinterpreted regarding the size of a VSM device and the duration of experiments. It is not properly a large-scale equipment, and the measurement time can be decreased to 5 min at room temperature. Yet, with the magnetic sensor, the measurement time is decreased to 10 s, a real convenience when testing time-dependent degradation effects on many samples. Also, to the best of our knowledge, a VSM equipment still costs in the order of 100-200k€, with also maintenance costs, while the magnetic sensor is ten times less expensive. Still, the magnetic sensor is definitely less powerful in terms of data it provides, being only a measure of a quantity of magnetic nanoparticles in the sample. We have thus totally rewritten the corresponding sentence to avoid any misleading interpretation.

“While standard magnetometry methods such as vibrating-sample magnetometer (VSM) provide direct information on the magnetic moment of a material as a function of temperature, field, and crystal orientation, the measurement time can be decreased only down to 5 min per sample at room temperature, and biological samples need to be fixed. Herein, with the magnetic sensor, the measurement time can be decreased to 10 s, a real convenience when testing time-dependent degradation effects on many samples, and to work with live biological samples. Yet, it only provides a measure of the quantity of magnetic nanoparticles in the sample, with principle introduced in more details in supplementary Figure S2 and in previous work. Besides, calibration (shown in Figure 1B) requires the use of VSM.”

Figure S2: Photos of the magnetic sensor device (A) and schematic describing the arrangement of coils to perform the frequency-dependent measure (B). In brief, this detection method is based on the nonlinear magnetization of magnetic sample, and consists in measuring this nonlinear response upon exposure to two-frequencies alternating magnetic field. The two-frequencies fields are generated by two independent coils, in a far range from 100 Hz to 100 kHz, and with different magnetic field amplitudes of 20 mT and 1 mT, respectively. The concept is to magnetize the nanoparticles with a low frequency field, and then switch this magnetization sinusoidally at high frequency so that the sensing coil voltage will be modulated by both frequencies. Combinatorial Fourier transform analysis finally provides the third derivative of the sample magnetization around zero field and room temperature. The most powerful aspect of this detection approach is that the combinatorial measurement gets rid of all background noise.

Response to Reviewer #3 (Remarks to the Author):

This paper titled “Photothermia at the nanoscale induces ferroptosis via nanoparticle degradation” is reporting that iron oxide nanoparticles can have a degradation with laser heating photothermia. During the study, global heating and local heating were defined for obtaining hot spot temperatures. Those correlation process logically performed and it is showing a meaningful results. Since there is a challenge to measure the hot spot temperature during thermoseed heating, those results and method are very interesting. Degradation dependent photothermia heating behavior of iron oxide nanoparticles can be observed in the different stage of nanoparticles cellular uptake too. Eventually, authors hypothesized that the photothermia triggered degradation will release Fe²⁺ to induce ferroptosis cell killing. Overall, the topic is interesting and research design/data of laser heating and degradation are well-presented. However, the main concern is the application part. The ferroptosis will be strongly associated with Fe²⁺ ion but the cell death mechanism is complex. Rigorous data that can prove the ferroptosis induction from the photothermia with nanoparticles is strongly required.

We thank the reviewer for the positive evaluation of our work and for the relevant comments that helped us improve the manuscript. We also acknowledge the main concern concerning the ferroptosis application, and have performed additional experiments to validate this claim, introduced in details in the answer to comment 5);

Additionally, some details are missing in the manuscript.

1) The developed magnetic sensor is not well described. Additional references, equation, and validation data will be helpful.

We have included a supplementary Figure S2 describing in more details the magnetic sensor, we have also briefly discussed its comparison with VSM technique, and have cited a reference that fully describes the device. We agree that this information was missing, and thank the reviewer for noticing.

“While standard magnetometry methods such as vibrating-sample magnetometer (VSM) provide direct information on the magnetic moment of a material as a function of temperature, field, and crystal orientation, the measurement time can be decreased only down to 5 min per sample at room temperature, and biological samples need to be fixed. Herein, with the magnetic sensor, the measurement time can be decreased to 10 s, a real convenience when testing time-dependent degradation effects on many samples, and to work with live biological samples. Yet, it only provides a measure of the quantity of magnetic nanoparticles in the sample, with principle introduced in more details in supplementary Figure S2 and in previous work. Besides, calibration (shown in Figure 1B) requires the use of VSM.”

Figure S2: Photos of the magnetic sensor device (A) and schematic describing the arrangement of coils to perform the frequency-dependent measure (B). In brief, this detection method is based on the nonlinear magnetization of magnetic sample, and consists in measuring this nonlinear response upon exposure to two-frequencies alternating magnetic field. The two-frequencies fields are generated by two independent coils, in a far range from 100 Hz to 100 kHz, and with different magnetic field amplitudes of 20 mT and 1 mT, respectively. The concept is to magnetize the nanoparticles with a low frequency field, and then switch this magnetization sinusoidally at high frequency so that the sensing coil voltage will be modulated by both frequencies. Combinatorial Fourier transform analysis finally provides the third derivative of the sample magnetization around zero field and room temperature. The most powerful aspect of this detection approach is that the combinatorial measurement gets rid of all background noise.

2) It is not clear the morphological change of nanoparticles after the photothermia degradation.

In cells, as shown in Figures S8 and S9 of the initially submitted SI file (now Figures S11 and S12 of the revised version), it is difficult to observe and quantify the degradation of nanoparticles. Besides, in other works (references 73,74), it has been shown that the kinetics of intracellular degradation with those same nanoparticles was so rapid no changes in the magnetic diameter (measured by magnetometry) could be detected, suggesting an “all-in-one” mechanism (once the nanoparticle start degrading, changes in diameter cannot be detected at the time scale of the measure).

For these reasons, we have also performed TEM imaging on nanoparticles after photothermal application, either in water (pH7, no degradation), or in degrading medium (pH4.5 with citrate), now included as a new supplementary Figure S13. While there are absolutely no changes in the nanoparticles size after laser application in water, a small, yet significant decrease is seen when in degrading medium. Nevertheless, in this case, most nanoparticles were fully degraded, thus not detected on TEM images.

This is now commented in the revised manuscript:

“TEM images of spheroids at early and late endosomal nanoparticle internalization and subsequent laser exposure are shown in supplementary Figures S11 and S12, respectively. It resembles the spontaneous degradation patterns of iron oxide nanoparticles observed in other works, without stimulation.^{73, 74} It is not possible to detect degraded nanoparticles on these images. However, some intact nanoparticles can still be detected, along with other structures such as iron-loaded ferritin, or dark fingerprints within endosomes, both likely a result of iron oxide nanoparticles degradation. In order to still have a quantitative evaluation of the laser-mediated effect on nanoparticles at the nanoscale, laser was applied either in water (pH7, no degradation), or in degrading medium (pH4.5 with citrate), and samples were observed with TEM, as shown in supplementary Figure S13. While there are absolutely no changes in the nanoparticles size after laser application in water, a small, yet significant decrease is seen in a degrading medium.”

The new supplementary Figure S13 inserted in the revised SI file is shown below:

Figure S13: TEM imaging of nanoparticles dispersed in water, without laser application (A,B), with laser application (C,D) and with laser application but at pH 4.5 supplemented with citrate (E,F). Laser was applied for 10 min at 3 W/cm² (60% degradation). Diameter distributions (B,D,F) were obtained for 400 measurement of individual nanoparticles. Average diameter was found to be 10.2±1.4 nm and 10.2±1.7 nm, in water, without and with laser application, respectively, and decreased significantly to 9.4±1.2 nm in degrading medium with laser application ($p < 0.0001$).

3) Is there any possibility of crystallinity change of iron oxide nanoparticles after photothermia? X-ray diffraction data will be helpful.

We thank the reviewer for the comment. XRD would give us information about the crystalline size but would not give us information about oxidation state of the iron oxide and changes from magnetite to maghemite structures. So we decided to perform FTIR experiments before and after laser exposure to get information about the iron oxidation state after photothermia. This is now included as new supplementary Figure S3. Laser irradiation in water does not affect neither the Fe-O vibration band within the 800-450 cm^{-1} region nor C-O stretches within the 1700-1300 cm^{-1} region. This is respectively correlated to unchanged coating surface and crystallinity.

Figure S3: FTIR spectra of citrate coated non oxidized (NP) before (blue curve) and after NIR laser irradiation (black curve). Laser irradiation in water does not affect neither the Fe-O vibration band within the 800-450 cm^{-1} region nor C-O stretches within the 1700-1300 cm^{-1} region. This is respectively correlated to unchanged coating surface and crystallinity.

4) Need more discussion on only heating effect on Fe²⁺ intracellular ions.

We have now performed laser exposure on a solution of ferrous ascorbate under the same experimental settings than the curves reported in Figure 2B for nanoparticles. It revealed almost no heating efficiency with only Fe²⁺ ions, as the results are now introduced in the new supplementary Figure S4.

Figure S4: Temperature increase of a solution of ferrous ascorbate at $[\text{Fe}] = 2 \text{ mM}$ in water exposed to increasing 808 nm laser power densities under the same experimental settings of the heating recorded with nanoparticles presented in Figure 2B. The temperature increases at 1.5, 2, 2.6, and 3 W/cm^2 were 3, 5, 7 and 9°C , compared to 22, 34, 44 and 58°C for ferrous ascorbate and iron oxide nanoparticles, respectively. Overall, Fe^{2+} provide less than 15% of the heating efficiency of nanoparticles.

5) Additional ferroptosis markers should be tested.

We have now included more data showing that ROS generation was triggered, as well as lipid peroxidation, as new supplementary Figures S18 and S19. Both are discussed in the revised version of the manuscript, as follows:

“Such an important release of Fe^{2+} could induce Fenton reaction in cancer cells and produce cytotoxic reactive oxygen species (ROS). To assess possible intracellular ROS generation, spheroids were laser irradiated and ROS were detected using the green fluorescent marker DCFH-DA. Typical images of the spheroids are shown in supplementary Figure S18A-D, indicating an important generation of ROS for the 1.6 W/cm^2 condition, that decreased for 2.1 W/cm^2 , and was almost zero for 2.6 W/cm^2 . Quantification of the relative fluorescent signal at single spheroid level ($n > 100$), shown in Figure S18E, confirms the massive ROS generation at low laser power decreasing with laser power. It shows that the released Fe^{2+} can indeed trigger ROS generation through the Fenton reaction, but only if the PTT effect is small enough to not kill the cells. By contrast, if PTT is sufficient to lead to complete cancer cell death, as for instance at 2.6 W/cm^2 (Figure 5G), ROS generation is drastically reduced.

Including the Fenton reaction with iron as a catalyst, the ferroptosis pathway is part of a complex cascade of intercellular reactions, where one reliable and well-studied read-out being lipid peroxidation.⁷⁵ We thus evaluated the production of lipid peroxides in laser-irradiated spheroids using the specific green fluorescent dye LiperFluo. Typical fluorescent images are shown for different laser power densities in supplementary Figure S19A-D, evidencing that the lipid peroxidase presence is significantly higher for the lower power density condition of 1.6 W/cm^2 , whereas it is almost non-existent for the higher power density of 2.6 W/cm^2 . Yet, lipid peroxidation was still more important after treatment with the ferroptosis-inducing agent Erastin (Figure S19E). The visual results were confirmed by quantitative analysis at the single-spheroid level (Figure S19G), overall suggesting the occurrence of ferroptosis for the low laser power density condition, akin to the ROS generation.

Similarly, the ferroptotic oxidative damage mechanism appears to be nullified at high power densities, where it is likely that the high local heating proves to be more efficient to induce cell death."

Figure S18: ROS generation in cells. In brief, spheroids were formed in microwells (using a 6-well plate stamp to generate 500 spheroids per well) from an initial seeding density of 500 cells labelled with nanoparticles, and matured over 48 hours. They were then transferred to a 96-well plate (100 per well) and exposed to laser with the same protocol as introduced in the Methods, at the three different power densities of 1.6, 2.1, and 2.6 W/cm², and for 5 min. 1 hour after exposure, the spheroids were washed with HBSS and incubated with green fluorescence ROS dye (DCFH-DA, Dojindo, R252) for 30 min according to supplier's instructions, washed again and observed with a plate reader (green channel). **A.** Typical images of the spheroids, indicating an important generation of ROS for the 1.6 W/cm² condition, that decreased for 2.1 W/cm², and was almost zero for 2.6 W/cm². **B.** Quantification of the relative fluorescent signal at single spheroid level (n>100) confirms the massive ROS generation at low laser power decreasing with laser power. *** p<0.0001.

Figure S19: Similarly to the previous ROS analysis setup, spheroids were irradiated at increasing laser power densities of 1.6, 2.1 and 2.6 W/cm² for a duration of 5 minutes. 3 hours later, the spheroids were incubated for 30 minutes with the fluorescent dye LiperFluo (Dojindo) according to supplier's instructions. Erastin (Sigma-Aldrich, E7781) was used as positive control to induce ferroptosis. Spheroids were then treated with Erastin (5μM) for 48 hours before incubation with LiperFluo. **A.** Typical fluorescent images of control, and the three laser conditions, and the Erastin condition. Lipid peroxidase presence is high for the lower power density condition (yet lower than with Erastin administration), whereas it is almost non-existent for the higher power density. **B.** Quantitative analyses of single spheroids relative fluorescence intensity (n>300). *** p<0.0001.

6) More discussion on *in vivo* application of the photothermia induced ferroptosis induction is essential.

This has now been discussed in the revised manuscript:

“Such a bimodal laser-mediated thermo-ferroptotic treatment certainly faces the same hurdles than photothermal therapy alone, with the most concerning one being the limited light body penetration. Even if near-infrared (NIR) wavelengths have a higher penetration depth over visible wavelengths, external NIR laser application can only penetrate a few mm in to deliver sufficient light for treatment. To envisage the thermo-ferroptotic cancer treatment *in vivo*, two options can be thought off. First, endoscopy can still be used to deliver light deeper, but only to tumors close to the lumen of the vessel. Second, while most PTT works (including this one) use laser in the first NIR windows (NIR-I, 700–900 nm), the second NIR window (NIR-II, 1000–1700 nm) might be considered for its even deeper transparency compared to NIR-I, associated with higher permissible exposure. More the less, iron oxide nanoparticles were shown to be efficient in the NIR-II.⁷⁶ NIR-II might thus be a better option to trigger laser-mediated ferroptosis *in vivo*. In addition, a major advantage of the dual thermo-ferroptotic treatment can be noted: the efficacy of ferroptosis is maximal at low laser power densities, whereas photothermal therapy is more efficient at higher nanoparticle concentrations, which are typically hard to reach if they are administered intravenously. As a matter of fact, this reflects the local heating at the nanoscale that we report in this work, which is not dependent on the nanoparticle concentration, and that triggers degradation and cytotoxic Fe²⁺ release. Thus, a remotely-triggered ferroptosis effect can prove advantageous over photothermal therapy under *in vivo* constraints, where irradiation is limited in its penetration, and nanoparticles accumulation can be low.”

7) There are some papers related with photo-initiated ferroptosis, caused by released ferrous ions from iron oxide. Authors may add more information that can represent the novelty of this study compared to the published reports.

We believe the main novelty of our work over existing ones to be the quantification of the release of cytotoxic Fe²⁺ from iron oxide nanoparticles upon laser exposure, by both magnetic measurements and Fe²⁺ imaging. Indeed, the magnetic measurements are the direct quantitative signature of the degradation of the nanoparticles, data that is scarcely provided. Besides, we also believe that we are the first to have used this magnetic quantification of degradation to determine the local heating in the nanoparticles vicinity, at the nanoscale, that we consider, as a matter of fact, the most impressive result of our work.

We have added this perspective in the conclusion:

“To the best of our knowledge, the work presented here offers the first insight into the quantification of the release of cytotoxic Fe²⁺ from iron oxide nanoparticles upon laser exposure, as well the first determination of the local temperature at the nanoparticle level, obtained from the magnetic signature of the nanoparticles upon degradation.”

Reviewers' Comments:

Reviewer #1:

Remarks to the Author:

The manuscript titled "Photothermia at the nanoscale induces ferroptosis via nanoparticle degradation" reports a magnetometry-based strategy to evidence the laser-induced hot-spot effect and its accelerating effect on magnetite nanoparticles' dissolution in real-time. The strategy and results provided in this study show certain significance in elucidating the mechanisms of magnetite nanomaterials-based hyperthermia/chemodynamic therapies. However, this paper is not suitable for publishing in Nature Communications which dedicates to providing high-quality research and important advances to specialists. The reasons are listed below:

1. Lack of novelty.

The methods, real-time in situ magnetic measurement of intracellular biodegradation of iron oxide nanoparticles by a benchtop-size magnetic sensor and the quantitative analysis, have already been reported by the same group elsewhere (Nano Research 2020, 13, 467–476).

The phenomena, such as "laser-induced hot-spot effect" and "hyperthermia-promoted dissolution of nanoparticles" are nearly common sense and have been reported by many previous publications (Small Methods, 2018, 2, 1800007; ACS Biomaterials Science & Engineering 2019, 5, 2, 1045–1056).

2. Some of the expressions in this manuscript are ambiguous.

For instance, laser irradiation alone could not induce the degradation of iron oxide nanoparticles, it works by triggering the photothermal effect that further promotes the dissolution of iron oxide nanoparticles in an acidic environment, as demonstrated in Figure 1 and Figure S3. However, there are many expressions like "laser induced/triggered nanoparticle degradation" in the manuscript, which should be rephrased to be more rigorous.

3. More detailed ferroptotic mechanism studies are suggested, as other signaling pathways may also induce lipid peroxidation.

Reviewer #2:

Remarks to the Author:

The paper has been properly revised and all the questions answered. In my opinion, the paper is now ready to be published as it is.

Reviewer #3:

Remarks to the Author:

The authors have adeptly addressed all of the concerns raised in the review.

Response to Reviewer #1: The manuscript titled “Photothermia at the nanoscale induces ferroptosis via nanoparticle degradation” reports a magnetometry-based strategy to evidence the laser-induced hot-spot effect and its accelerating effect on magnetite nanoparticles’ dissolution in real-time. The strategy and results provided in this study show certain significance in elucidating the mechanisms of magnetite nanomaterials-based hyperthermia/chemodynamic therapies. However, this paper is not suitable for publishing in Nature Communications which dedicates to providing high-quality research and important advances to specialists. The reasons are listed below:

We thank the reviewer for the new evaluation of our work. We provide below a point-by-point response to the new concerns.

- 1) Lack of novelty. The methods, real-time in situ magnetic measurement of intracellular biodegradation of iron oxide nanoparticles by a benchtop-size magnetic sensor and the quantitative analysis, have already been reported by the same group elsewhere (Nano Research 2020, 13, 467–476). The phenomena, such as “laser-induced hot-spot effect” and “hyperthermia-promoted dissolution of nanoparticles” are nearly common sense and have been reported by many previous publications (Small Methods, 2018, 2, 1800007; ACS Biomaterials Science & Engineering 2019, 5, 2, 1045–1056).

Authors’ response: The reviewer is correct in pointing out that the magnetic moment measurement methodology has recently been reported by us, as was rightly referenced in the manuscript (reference no. 53). However, we have not claimed in this new work that the methodology is of a novel nature, or that it is one of the main scientific advances being put forward herein.

Similarly, we have not claimed to evidence the first instance of a laser-induced hot-spot effect, or that of the hyperthermia-promoted dissolution of nanoparticles.

In fact, taking advantage of the previously reported existing methodology, we set out to assess what we believe are the main novel aspects of our work:

1. The evidence of the mechanistic relationship between temperature increase and nanoparticle degradation.
2. An alternative estimation of the local “hot-spot” temperature, inferred in a rather simple and intuitive way from the nanoparticles’ degradation profile.
3. The dependance of the induced nanoparticle degradation on the endocytosis state.
4. The dependence of the resulting ferroptosis response on the irradiation intensity, i.e. the temperature increase, as well as on the oxidation state of the nanoparticles.

The referenced article (ACS Biomaterials Science & Engineering 2019, 5, 2, 1045–1056) explores the release of iron ions due to increases in temperature only in solution, analyzed both by ICP-MS and methylene blue bleaching. Then, they show *in vitro* the viability of 2D culture cells after photothermal irradiation, as well as the presence of ROS. However, no resulting ferroptosis evidence is provided.

In contrast, here we believe to provide to the community the first evaluation of the tight interplay between photothermia, nanoparticle degradation and the resulting ferroptotic response in a relevant, high-throughput 3D tumor cell culture model.

- 2) Some of the expressions in this manuscript are ambiguous. For instance, laser irradiation alone could not induce the degradation of iron oxide nanoparticles, it works by triggering the photothermal effect that further promotes the dissolution of iron oxide nanoparticles in an acidic environment, as demonstrated in Figure 1 and Figure S3. However, there are many expressions like "laser induced/triggered nanoparticle degradation" in the manuscript, which should be rephrased to be more rigorous.

Authors' response: We have now changed these phrases throughout the manuscript to reflect the actual photothermal effect indicated. We hope the changes are found to be satisfactory.

- 3) More detailed ferroptotic mechanism studies are suggested, as other signaling pathways may also induce lipid peroxidation.

Authors' response: While there certainly are various signaling pathways that culminate in lipid peroxidation, they typically require a certain stress or injury to be activated. As indicated in existing literature on lipid peroxidation (Biochemical and Biophysical Research Communications 2017, 482, 3, 419-425; Oxidative Medicine and Cellular Longevity, 2014, 360438), the non-enzymatic lipid peroxidation pathway is initiated by the iron-dependent Fenton reaction, whereas the enzymatic pathway is regulated mostly by lipoxygenases, which is in turn iron-dependent. Moreover, lipoxygenase activity can further be regulated as a response to stresses such as osmotic shock, ultraviolet and ionization irradiations, and to certain diseases and infections.

In this work, considering that the only external stress sources inherent to our system or therein introduced are the iron oxide nanoparticles and a photothermal increase in temperature, we do not expect the activation of pathways outside of these domains to take place. Regarding the possibility of heat shock from photothermal irradiation as another route leading to the reported lipid peroxidation, there are at least two obvious reasons why it is unlikely to be the trigger. Firstly, we observed the highest rate of lipid peroxidation at lower laser power density (Figure S19). Increasing the laser power density resulted in a lower rate of lipid peroxidation, suggesting that higher temperature conditions, such as those in therapeutic settings, do not induce lipid peroxidation. However, in the re-revised manuscript, we provide a second piece of evidence by detecting lipid peroxidation following photothermal irradiation of oxidized nanoparticles. Remarkably, we found a significantly reduced level of lipid peroxidation (**new Figure S20**) under the same photothermal irradiation settings. As we demonstrated in the initial manuscript, the use of oxidized nanoparticles (conversion of ferrous ions into ferric ions) resulted in minimal

amounts of Fe²⁺ released intracellularly (Figure 5I). This thus provides direct evidence that the reported lipid peroxidation is associated with the presence of Fe²⁺.

Therefore we present compelling evidence of a strong correlation between ROS, presence of Fe²⁺ and lipid peroxidation. This tight correlation serves already as a strong indication of ferroptosis activation, akin to that elicited by the ferroptotic agent Erastin, which we also included as positive control.

Furthermore, in the re-revised manuscript, we also present the expression analysis of two genes SLC7A11 and CHAC1, which are known to be involved in ferroptosis (**new Figure S21**). The results reveal that photothermal irradiation leads to a decrease in intracellular glutathione levels by reducing its import (downregulation of SLC7A11) and promoting its degradation (upregulation of CHAC1), thereby triggering ferroptosis. Importantly, we also observe the upregulation of CHAC1 following treatment with Erastin, confirming other studies showing that CHAC1 is one of the genes predominantly overexpressed in Erastin-induced ferroptosis.

Response to Reviewer #2: The paper has been properly revised and all the questions answered. In my opinion, the paper is now ready to be published as it is.

We thank the reviewer for the positive evaluation of the work.

Response to Reviewer #3: The authors have adeptly addressed all of the concerns raised in the review.

We thank the reviewer for the positive evaluation of the work.

Reviewers' Comments:

Reviewer #1:

Remarks to the Author:

The manuscript has been properly revised and all the questions have been answered.